# BENCHMARKING DIVERSITY IN IMAGE GENERATION VIA ATTRIBUTE-CONDITIONAL HUMAN EVALUATION

## ABSTRACT

Despite advances in generation quality, current text-to-image (T2I) models often lack diversity, generating homogeneous outputs. This work introduces a framework to address the need for robust diversity evaluation in T2I models. Our framework systematically assesses diversity by evaluating individual concepts and their relevant factors of variation. Key contributions include: (1) a novel human evaluation template for nuanced diversity assessment; (2) a curated prompt set covering diverse concepts with their identified factors of variation (e.g. prompt: *An image of an apple*, factor of variation: color); and (3) a methodology for comparing models in terms of human annotations via binomial tests. Furthermore, we rigorously compare various image embeddings for diversity measurement. Notably, our principled approach enables ranking of T2I models by diversity, identifying categories where they particularly struggle. This research offers a robust methodology and insights, paving the way for improvements in T2I model diversity and metric development.

## 1 MEASURING DIVERSITY IN TEXT-TO-IMAGE MODELS

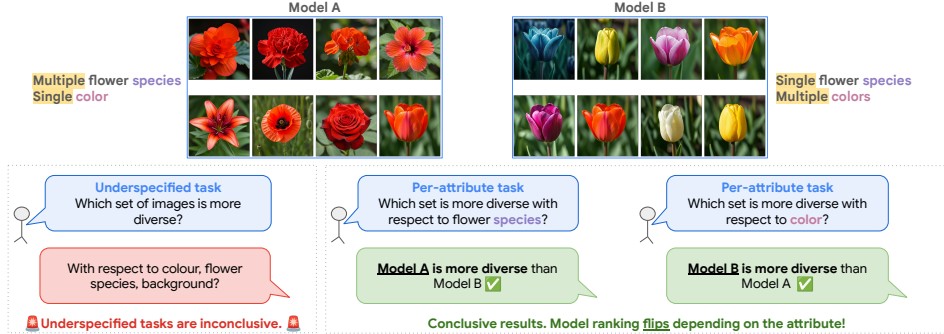

Figure 1: Evaluating diversity requires specifying both the concept being assessed and the factor of variation to reduce ambiguity in the annotation process.

Output diversity is widely considered desirable for text-to-image (T2I) generation models aiming to accurately represent the natural variability of entities in the real world. This is crucial not only technically, for serving as faithful world models, but also for downstream applications like supporting creative processes and ensuring broad conceptual representation across contexts. For example, a diverse model generating "an image of a house" should produce variations in architectural style and background. However, current diversity metrics often conflate it with other properties like fidelity (e.g., Fréchet Inception Distance (FID) (Heusel et al., 2017)). While progress has been made by developing dedicated metrics (e.g., Vendi Score (Friedman & Dieng, 2022)), the conditions for measuring diversity remain poorly defined and lack standardization, highlighting the need for a principled framework.

In particular, previous work often measures the variability of generated images in scenarios that do not explicitly account for diversity. For instance, images may be generated using a prompt set that neither requires nor controls for output variations (e.g., Sadat et al., 2024; Astolfi et al., 2024), or models may be compared using a generic human evaluation template that does not specifically probe for diversity (e.g., Betker et al., 2023). This can result in measures of diversity that are ambiguous or

inconclusive (see Fig. 1). To address this challenge, we propose a framework to measure diversity without conflating constructs (Zhao et al., 2024a;b; Mironov & Prokhorenkova, 2024; Jalali et al., 2024; Vrijenhoek et al., 2024): we operate under the premise that systematically evaluating diversity requires specifying both the concept being assessed and the attribute of interest, as illustrated in Fig.1. We empirically validate this by demonstrating that human accuracy in evaluating diversity is at chance level when the attribute is not defined. Building on this observation, we introduce a novel evaluation framework designed to measure the per-attribute intrinsic diversity of T2I models. This framework includes a synthetically generated prompt set spanning common concepts and their variations, as well as a human evaluation template. The template, informed by empirical findings on a golden set, improves human accuracy by dividing the evaluation into two subtasks: counting and counts comparison.

Considering the high cost of human evaluations for model ranking, developing automated metrics that accurately reflect human judgment is crucial for advancing T2I models. While various diversity metrics have been proposed (Friedman & Dieng, 2022; Jalali et al., 2024), their alignment with human perceptions of diversity often remains unevaluated. To address this, we use our proposed human evaluation template and prompt set to examine the reliability of autoevaluation metrics. Specifically, we investigate the Vendi Score (Friedman & Dieng, 2022), a widely adopted diversity metric (Kannen et al., 2024b; Hemmat et al., 2024) whose correlation with human-perceived diversity has not yet been thoroughly established. Our analysis reveals that the Vendi Score, when optimized for the appropriate representation space, can achieve approximately 65% accuracy in capturing human diversity judgments. We also find that the accuracy improves to 80% when the model pairs are more different, highlighting the need for more discriminant representations. Furthermore, we apply our framework to compare five recent generative models: Imagen 3 (Baldridge et al., 2024), Imagen 2.5 (Vasconcelos et al., 2024), Muse 2.2 (Chang et al., 2023), DALLE3 (Betker et al., 2023), and Flux 1.1 (Labs, 2024). This comparison identifies Imagen 3 and Flux 1.1 as the top-performing models regarding attribute diversity. We believe our framework provides a robust foundation for future work in developing more human-aligned evaluation metrics and improving T2I model diversity. This research makes three key contributions:

- We formalize the problem of quantifying diversity in T2I models and introduce a practical evaluation framework based on pre-defined factors of variation.
- We introduces an evaluation framework consisting of the first human evaluation template tailored for diversity, a prompt set covering 86 concept-factor variation pairs, and statistical hypothesis test to compare models.
- We use the proposed framework to collect a comprehensive dataset of 24591 human annotations comparing 5 prominent T2I models and use this data to rank automatic evaluation metrics. Prompts are available in the Supplementary Material and the full benchmark (annotations, images, and prompts) will be released upon publication.

## 2   THE THREE INGREDIENTS FOR DIVERSITY EVALUATION

To evaluate diversity, our framework is based on three components: a definition of what specific diversity is being measured, a prompt set to elicit relevant outputs, and a human evaluation template for reliably comparing models. These are described below.

### 2.1   A CLEARLY SPECIFIED PROBLEM: DIVERSITY PER ATTRIBUTE

**Prelude: formalizing diversity.** Consider a set of images $X = \{x_1, x_2, \ldots, x_n\}$, where each image $x_i$ belongs to a space $\mathcal{X} \subseteq \mathbb{R}^D$. We posit that the visual appearance of each image $x_i$ is primarily determined by a set of $K$ underlying independent generative factors $f_i = \{f_i^1, \ldots, f_i^K\}$. A potential generative model could be formulated as:

$$p(x_i) = \prod_{k=1}^{K} p(x_i|f_i^k)p(f_i^k). \tag{1}$$

We focus on scenarios where images represent scenes containing instances from well-defined concepts (e.g., bottle, forest). Given a concept, we can often map these abstract generative factors to concrete, observable attributes. For instance, an image $x_i$ depicting a bottle can be described by attributes such as: $f^{\text{material}} \in \{\text{glass, plastic, metal}\}$, $f^{\text{shape}} \in \{\text{cylindrical, square}\}$, and $f^{\text{state}} \in \{\text{open, closed}\}$.

Let $C = \{c^1, \ldots, c^J\}$ be the set of concepts, $A^j = \{a^{j,1}, \ldots, a^{j,K}\}$ the relevant attributes for a given concept $c^j$, and $V^{j,k}$ the finite set of possible values for attribute $a^{j,k}$. Each image $x_i$ depicting a concept is associated with a specific value $v_i^{j,k} \in V^{j,k}$ for each attribute $a^{j,k}$. We define a sample of images $X^j$ (for the same concept $c^j$) as *perfectly diverse* if it comprehensively covers all attribute variations. More precisely, for every attribute $a^{j,k} \in A^j$ and every possible value $v \in V^{j,k}$ there must exist at least one image $x_i^j \in X^j$ such that the attribute $a^{j,k}$ for image $x_i^j$ takes the value $v$.

**A tractable notion of diversity.** Measuring diversity across the complete set of generative factors underlying natural data is significantly challenging. Firstly, the sheer number of potential factors ($K$) is often immense. Secondly, as highlighted by Tsirigotis et al. (2024), the combination of their possible values grows exponentially, leading to a 'curse of generative dimensionality' where no realistic finite sample can cover all possible combinations. Thirdly, many factors may inherently possess continuous value ranges, making exhaustive coverage impossible even for a single factor.

Given these challenges, and since achieving the *perfect diversity* (as defined earlier) is intractable with a finite sample, we instead propose to measure *tractable diversity*. This approach focuses on a carefully selected subset of the most salient and practically relevant generative factors ($K'$) for a specific concept. Identifying which factors are practically relevant is non-trivial and must be tailored for a given use case. In this work, to identify these factors, we focus on commonly observed concepts reflective of T2I model training data. To effectively sample from the distribution of generative factors within these concepts, we leverage the knowledge encoded by Large Language Models (LLMs) (Rassin et al., 2024). Specifically, we prompt an LLM (Gemini 1.5 M (Team et al., 2024)) to identify relevant aspects of variation for evaluating the diversity of a given concept. The full system instruction is given in the Appendix.

## 2.2 A SYSTEMATICALLY GENERATED PROMPT SET

Our goal is to rigorously evaluate generative models and diversity metrics, specifically focusing on their ability to represent variation within distinct attributes of concepts. To effectively rank these models and metrics, our framework must accommodate both precisely controlled scenarios and complex, real-world use cases. We deliberately select concepts that are ubiquitous in everyday life and common image datasets, such as ImageNet (Deng et al., 2009) (e.g., 'fruit', 'car', 'snake'), thereby anchoring our evaluation in practical utility. However, simple concepts alone are insufficient. They must also possess inherent complexity and variability, presenting a genuine challenge to the models and metrics. The chosen concepts and their attributes need to be sufficiently nuanced to allow our methodology to clearly reveal performance differences and track improvements over time or across different systems.

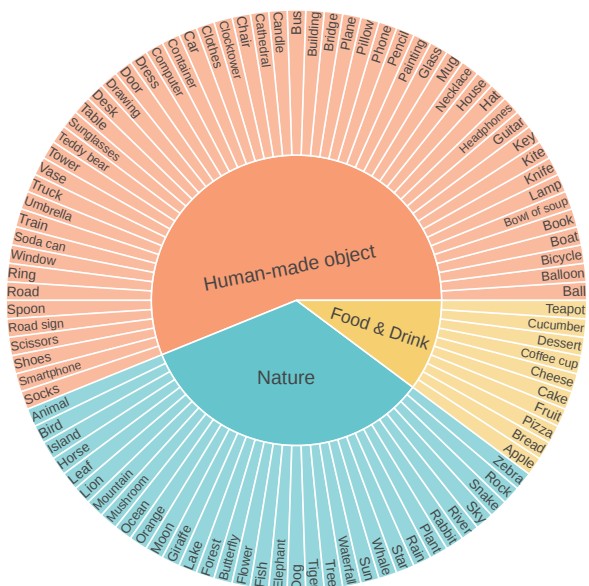

Figure 2: Each slice represents a concept, grouped and color-coded by its overall category.

To structure this process, we classify concepts into three widely applicable categories: *Food and Drink* (e.g. *coffee cup*, *cake*), *Nature* (elements e.g. *river*, *butterfly*), and *Human-made Objects* (e.g. *bridge*, *laptop*). We leverage the generative capabilities of Large Language Models (LLMs) to systematically produce a wide range of concepts within these categories, producing concrete, "ImageNet-like" concepts, which are typically visualizable nouns, similar in scope to those in large-scale image datasets. For each generated concept, the LLM is used to identify a semantically relevant aspect of variation (attribute) that is intrinsic or commonly associated with that concept. This yields concept-attribute pairs $(c^j, a^{j,k})$ such as: (*apple*, *color*), (*tree*, *species*),

(*coffee cup*, *material*), (*chair*, *style*). This LLM-driven process allows us to systematically build a prompt set specifically designed to probe and evaluate diversity along meaningful, contextually relevant dimensions for a broad range of common concepts. Finally, the authors manually verified all concept-attribute pairs and removed 5 where the attribute was potentially difficult / ambiguous to categorize (e.g. (*food*, *cuisine*)). The specific prompt used can be found in Appendix D.1. Additionally, in Appendix D.2 we discuss the sufficiency of our prompt set to discriminate models.

## 2.3 A VALIDATED, BESPOKE HUMAN EVALUATION TEMPLATE

Prior work has shown that developing an appropriate human evaluation template is an essential component in the process of measuring a desired capability of a generative model (Wiles et al., 2024; Clark et al., 2021). To that end, we develop a human evaluation template that: (a) allows annotators to understand the task well, (b) captures their judgment faithfully, and (c) yields meaningful ground truth annotations for per-attribute diversity, subsequently used to validate automated evaluation metrics. The annotators are provided with 4 options for the side-by-side comparison: (i) Left more diverse, (ii) Right more diverse, (iii) Equally diverse, (iv) Unable to answer. Visualizations of the template can be seen in Appendix B.2.

**A template to measure per-attribute diversity.** Our template for measuring per-attribute diversity employs a comparative, side-by-side approach due to the difficulty of evaluating diversity within a single set. Many existing diversity metrics also require a reference set. We considered the following design choices for our human evaluation template to ensure meaningful assessment (1) *Set size*: Balancing the perception of diversity with minimizing annotation fatigue and enabling robust computation for metrics requiring larger sets (e.g., Vendi score). (2) *Attribute specification*: Explicitly stating the attribute for evaluation versus allowing open-ended diversity assessment. (3) *Anchoring task*: Incorporating an intermediate task to guide annotators to focus on the intended attribute.

**Validating the template with a golden set.** To evaluate the quality of the evaluation template, we curate a golden set of 10 `<concept, aspect>` pairs, where `concept` corresponds to a concept that should be considered common across images in a set and `aspect` describes the associated aspect of variation that we want to measure diversity against. The full list of concepts and aspects of variation can be found in Appendix B.1. We validate the evaluation template by comparing cases where (i) the concept *remains constant* across images in the set while the aspect *varies* (ii) the concept *varies* across images while the aspect *remains the same*, and (iii) *both* the concept and the aspect vary across images within the set. We expect images in set (i) to be considered more diverse than images in set (ii), and similarly images in set (iii) to be considered more diverse than images in set (ii). Finally, we expect that images in sets (ii) and (iii) are considered equally diverse as we want to focus on the `aspect` as axis of variation.

In Fig. 3, we present the annotation accuracy of human experts using our template under various conditions, considering the aforementioned definitions as ground truth. The different templates are shown in Fig. 9. The accuracy for the `w/o aspect` task is 30.0% for comparisons of sets of size 4 and 26.7% for sets of size 8. In contrast, the template that includes the `aspect` shows a significant increase in accuracy (82.5% for set size 4 and 53.3% for set size 8), indicating that explicitly mentioning the desired aspect of variation improves accuracy. This improvement likely stems from preventing annotators from unintentionally conflating the `concept` and the `aspect` when not guided to focus on a specific axis. Furthermore, we observe that adding the `count`

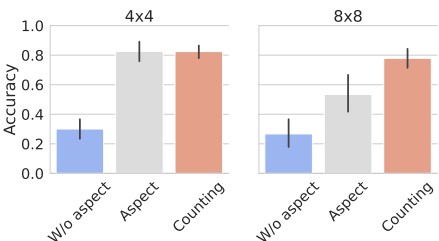

Figure 3: Match with the golden set depending on different set sizes.

anchoring question enhances accuracy, especially for the set size of 8, reaching 77.9%.

For the `count` task, we found a strong ($\rho = 0.88$) and statistically significant ($p < .001$) correlation between the annotators' final diversity comparison and the comparison inferred from their individual subset counts (where a higher count on one side implies a `more diverse` final response for that side, and equal counts imply `equal` diversity). This confirms that the `anchoring` count question effectively guides annotators. To further validate our setup, we analyzed instances where annotators' responses deviated from the ground truth in our golden set. We examined the distributions of attribute

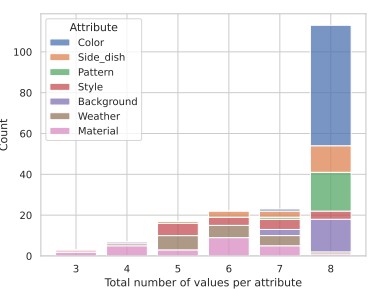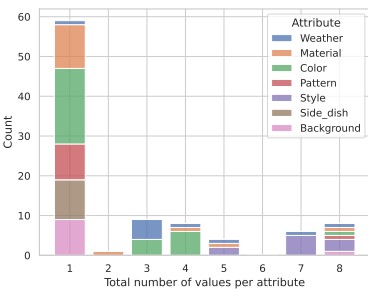

(a) The "diverse" golden set.       (b) The "non-diverse" golden set.

Figure 4: The distribution of counts for sets of images labelled as "diverse" or "non-diverse" in the golden set for the pilot study.

counts for two image subsets: (1) those labelled "diverse" in the ground truth, where we expected a count mode of "8" and (2) those labelled "non-diverse", where we expected a mode of "1". The results of this analysis are presented in Fig. 4. While generally, annotator responses aligned with the golden set labels, we observed a few exceptions. For instance, in one case labelled as a diverse set of chairs, all annotators counted only 3 or 4 distinct chair types, indicating lower diversity than expected. Upon closer inspection, these chairs appeared visually similar despite potentially different underlying material prompts (e.g., metal, iron, aluminum).

## 3    OUR FRAMEWORK IN PRACTICE

We demonstrate our framework's practical application by: (i) collecting comprehensive human annotations with our template to compare models, (ii) using these annotations as ground truth to evaluate diversity metrics, and (iii) comparing model rankings from human versus automatic evaluations to highlight the gap between human-perceived diversity and current metric capabilities.

### 3.1    RANKING MODELS VIA HUMAN EVALUATION

With the proposed prompt set from Sec. 2.2 and the human evaluation template introduced in Sec. 2.3, we evaluate the attribute-based diversity of five generative models, namely: Muse 2.2 (Chang et al., 2023), Imagen 2.5 (Vasconcelos et al., 2024), Imagen 3 (Baldridge et al., 2024), DALLE3 (Betker et al., 2023), and Flux 1.1 (Labs, 2024). For each model, we generate 20 distinct samples for each prompt, randomly combine them in 10 different sets of 8 images, and run side-by-side evaluations for all 10 combinations of 2 models. For each side-by-side comparison, evaluations from 5 different raters were collected. Raters had access to a slide deck with instructions to perform the task and were compensated for the time invested in the data collection. In total, 24591 annotations were collected in our study from 20 different annotators, including the pilot runs. The average time to complete the task with the final template was 32 seconds More details can be found in the Appendix (Sec.A). Before comparing each model pair in terms of diversity, we evaluate the overall annotations quality by computing the inter-annotator agreement via Krippendorff's alpha reliability ($\alpha$) (Hayes & Krippendorff, 2007). In Fig. 5a, we observe that for all cases $\alpha > 0.8$, indicating a high-degree of agreement across annotators (Marzi et al., 2024).

**Ratings aggregation.** Given the high levels of inter-annotator agreement for all runs of the human evaluation, we aggregate annotations for each side-by-side comparison across raters by *taking the mode* of the ratings. We then follow this step with a second aggregation, this time at the level of all side-by-side comparisons for each concept. For instance, when comparing a given model pair, there are 10 side-by-side comparisons for the concept *apple* (each side-by-side comparison here corresponds to the evaluation of two sets of 8 images). At the end of this process, for the considered models pair, we obtain a single human evaluation result for each concept in the prompt set.

**Model ranking.** Using the results from the ratings aggregation, we propose to use Binomial tests to verify the following hypothesis: *there is a significant difference between the outcomes of a given pair of models*. To do so, we count the number of categories for which each model was deemed best and perform a two-sided Binomial test under the null-hypothesis that the rate for which each model is

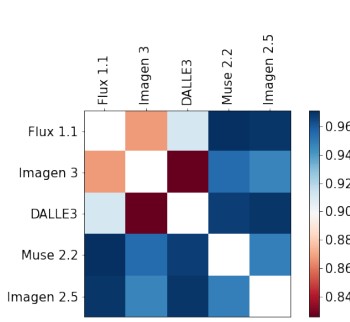 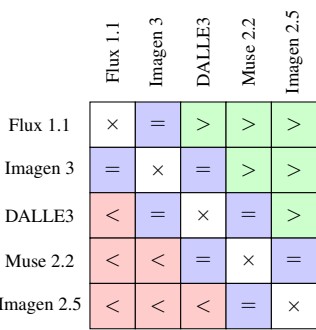

(a) Krippendorff's $\alpha$-reliability.    (b) Binomial test results at 95% confidence.

Figure 5: **Human evaluation results.** (a) Inter-annotator agreement results in terms of Krippendorff's $\alpha$-reliability. (b) We compare model rankings in terms of significance in the number of wins with two-sided Binomial tests under a 95% confidence level. Each entry in the grid represents a comparison between two models. The sign indicates the model in the row is better ($>$), worse ($<$), or not significantly different ($=$) than the model in the column.

the best for a concept is equal to 50% (i.e. both models have equal win rates). Results considering a 95% confidence level for all tests are shown is Fig. 5b. Imagen 3 and Flux 1.1 are significantly better or not worse than all other models. Imagen 2.5 and Muse 2.2 are not significantly better than any contender, showing that our benchmark is able to capture an overall progress in diversity when comparing newer and older models. DALLE3 is significantly better than Imagen 2.5, but does not significantly surpass the performance of the other models considered for comparison.

## 3.2 COMPARING AUTOEVALUATION METRICS

While human evaluation is often considered gold standard, it can be impractical to rely solely on human annotation. We then leverage the collected human annotations to perform an extensive study of the role of embeddings for the Vendi Score [1].

**Autoraters based on the Vendi Score.** Given a set of images $X^{j,k} = \{x_i^{j,k}\}$ (corresponding to a given model, concept $c^j$ and attribute $a^{j,k} \in A^j$), we extract embeddings $h_\Xi(x_i^{j,k})$ for each image. $h_\Xi$ is a pretrained feature extractor that can be dependent on a set of conditions $\Xi = \{\xi_l\} \subset (C \times A) \cup \{\xi^0\}$ where $\xi^0$ is a condition unrelated to the considered categories and attributes that can be added to test the impact of conditioning. The different feature extractors and conditions we used are detailed in the following paragraph, but here are a few generic examples to clarify the notation: (i) $h_\Xi$ takes only images as input. In this case, $\Xi = \emptyset$. (ii) $h_\Xi$ is a vision and language model. In this case, embeddings can be conditioned on text data that depends on the concept only (i.e., $\Xi = \{c^j\}$), attribute only (i.e., $\Xi = \{a^{j,k}\}$), or both concept and attribute (i.e., $\Xi = \{c^j, a^{j,k}\}$). To test the impact of conditioning on text, we can instead choose an unrelated prompt (i.e., using $\Xi = \{\xi^0\}$). Finally, we aggregate the embeddings using a diversity metric to obtain a score for the set. As we do not have access to a reliable reference in our setting, we use the Vendi Score (Friedman & Dieng, 2022), a reference-free and widely adopted metric (Pasarkar & Dieng, 2023; Jalali et al., 2024; Hemmat et al., 2024; Kannen et al., 2024a). The Vendi Score is defined as follows:

**Definition 1** (Adapted from (Friedman & Dieng, 2022), Definition 3.1). *Given a concept $c^j$, an attribute $a^{j,k}$ and a set of conditions $\Xi$, let $\{x_1^{j,k}, \ldots, x_n^{j,k}\}$ denote a set of images representing a given concept and attribute. Let $k: X \times X \to \mathbb{R}$ be the cosine similarity between the embeddings of two images, $K^\Xi \in \mathbb{R}^{n \times n}$ be the kernel matrix, with $K_{lm}^\Xi = k^\Xi(x_l^{j,k}, x_m^{j,k})$, and let $\lambda_1^\Xi, \ldots, \lambda_n^\Xi$ be the eigenvalues of $K^\Xi/n$. The Vendi Score for the set $\{x_1^{j,k}, \ldots, x_n^{j,k}\}$ is defined as:*

$$s_\Xi(x_1^{j,k}, \ldots, x_n^{j,k}) = \exp(-\sum_{i=1}^{n} \lambda_i^\Xi \log \lambda_i^\Xi). \tag{2}$$

---

[1]Results with other autoraters can be found in the Appendix Sec.E.

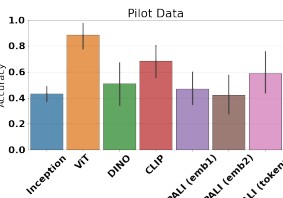 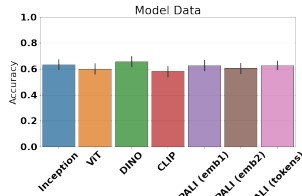 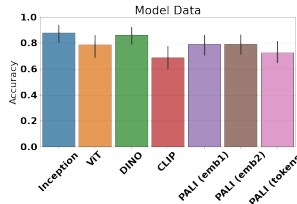

(a) The "diverse" golden set.

(b) Side-by-side model comparisons.

(c) Side-by-side model comparisons with diversity gap $> 4$.

Figure 6: **Autoevaluation results:** the performance of the Vendi Score given different embeddings across three settings: (a) the golden set; (b) all the annotations gathered; (c) the "easy" subset of the annotations where raters identified a diversity gap of $> 4$ for a pair. On the golden set, VIT performs best but this does not transfer to side-by-side comparisons. The performance is generally better on the "easy" split of the data, showing that the embeddings perform considerably worse when the difference between the generated sets of images is more subtle—models are more similar.

**Experimental setup.** We compare three different types of embeddings. First, we compare embeddings obtained *using only* the image input. Here we consider two models trained for IMAGENET classification – the IMAGENET INCEPTION model introduced in Szegedy et al. (2015) and an IMA-GENET VIT-B/16 model trained on IMAGENET21K as described in Steiner et al. (2022). We also consider one self-supervised model, DINOV2 (Oquab et al., 2023). Second, we consider embeddings conditioned on both the image and textual attribute. We use PALI embeddings Beyer et al. (2024) at various points after fusing the text and visual input, and CLIP (Radford et al., 2021) combined text and image embedding. We use these embedding models to obtain an embedding for each image in a set the Vendi Score in order to aggregate embeddings and obtain a diversity prediction for the set. Finally, we consider the first word output by the PALI model as a discrete token. We aggregate these outputs by counting the number of unique words generated for a set to get an estimate for diversity. For each pair of image sets, we analyze the agreement between a diversity assessment based on our autoraters, and the assessment resulting from the human annotations, not taking into account pairs where the annotators found the sets to be equally diverse. If the autoraters and the human evaluations both indicate the same set as being the most diverse (i.e., $s_\Xi(X_1^{j,k}) > s_\Xi(X_2^{j,k})$ and annotators rated the set $X_1^{j,k}$ generated with model 1 based on concept $c^j$ and attribute $a^{j,k}$ as more diverse than $X_2^{j,k}$ generated with model 2 based on the same concept and attribute), we say that for that pair of sets, the autorater is correct, else it is incorrect. We then report accuracy by aggregating the number of pairs for which the autoraters are correct.

**Results.** Results are reported in Figs. 6a-6c. We can see that, on the "diverse" golden set, the VIT model does the best, and then the tokens of PALI. This is perhaps surprising, as the VIT model is not specifically trained to focus on the aspects we are considering for diversity but to be able to discriminate between broad classes. However, we see minimal difference in results if we consider the model data. All approaches perform similarly and lead to accuracies that are not significantly different. We hypothesize that the reason for the observed small difference in results was that the models were similar to each other. As a result, we looked at ratings where the annotators perceived a larger gap between models by using the counts as a proxy. We consider a subset of the data where the difference in counts between the two sets is greater than 4, keeping about $24\%$ of the data. We find that now, on the model data we see a bigger difference in results. First, all autoraters are more accurate. Second, we can see that again the image based approaches (e.g., the INCEPTION model, the DINO model and VIT model) perform best. In Sec. E.3, we provide qualitative results showing which sets different embeddings deem as high or low diversity.

### 3.3 EVALUATING DIVERSITY USING FOUNDATION MODELS

We leverage the power of multimodal LLMs such as the Gemini model family (Team et al., 2024) to assess whether they can be a competitive alternative to automatic metrics that rely on embeddings. We design a system instruction aiming to prompt the model to perform a two-step evaluation akin to the human evaluation task. The full instruction can be found in Sec. E.6. We evaluate these evaluators on the golden set and present the results on Fig. 7a. Gemini v2.5 Flash achieves the best

performance, surpassing human accuracy in the task. A closer look at the results reveals that both human and auto raters perform similarly in almost all the cases, with the mismatches corresponding to the evaluating of diversity for the pair <building, style>. We hypothesize judging diversity of architectural styles is a complex task that heavily depends on the cultural background of annotators, thereby being more accurately performed by a powerful vision-language models. We also evaluate how the best evaluator performs on predicting human annotations. In Fig.7b, we see that for Imagen 3 comparisons, the foundation model-based evaluator presents competitive performance in comparison to embedding-based automatic metrics, without relying on embeddings, although currently more costly as evaluating each pair requires a query from the foundation model.

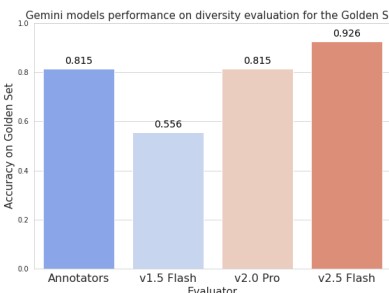 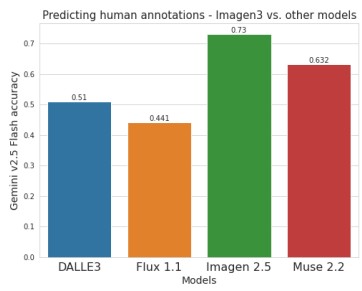

(a) Accuracy of autoraters based on the Gemini model family on the golden set.

(b) Gemini v2.5 Flash human evaluation prediction accuracy for Imagen 3 comparisons.

Figure 7: Evaluating diversity with Gemini. Gemini v2.5 Flash achieves the highest accuracy on the golden set and is competitive with embedding-based metrics when predicting human annotations.

### 3.4 RANKING MODELS WITH AUTOEVALUATION APPROACHES

Ranking is achieved by counting the frequency at which the left model (model 1) achieves a higher score than the model on the top (model 2), i.e. we count how many times $s_\Xi(X_1^{j,k}) > s_\Xi(X_2^{j,k})$, with $X_1^{j,k}$ generated with model 1, and $X_2^{j,k}$ generated with model 2, and subtracting 0.5. More results can be found in Sec. E.5. In order to test the significance, we aggregate the scores per concept and perform a Wilcoxon signed-rank test under a 95% confidence level. In Fig.8a, we consider the ImageNet Inception embeddings, as they yielded the highest accuracy on the model data. In Figs.8b and 8c, we consider text-conditioned embeddings, as they are closest to our human evaluation procedure. We show the results using PALI(EMB1), as they show a marginal advantage on model data. On the middle panel, we show the results corresponding to conditioning the embedding model on the attribute only, while on the right panel, conditioning takes into account both attribute and object. Results with other embeddings can be found in (Sec. E.5). Through the autoevaluation model ranking, we find that independently of the chosen embedding, Imagen 3 is not worse than all other models, and Flux 1.1, Imagen 3 and DALLE3 are better than Imagen 2.5 and Muse 2.2. We also observe that using ImageNet Inception embeddings and PALI(EMB1) with a conditioning on object and attribute captures more differences across the 3 top models, and that using both types of the PALI(EMB1) embeddings captures more differences between Imagen 2.5 and Muse 2.2. By adopting the model comparison results obtained with the human annotations as shown Fig. 5b as ground-truth, we find that all used embeddings are of similar quality in terms of closeness to human perception of diversity. They all did not flip conclusions, but the autoevaluation approach seems more sensitive to certain variations depending on the choice of embedding model and conditioning. Text conditioning, while closest to the human evaluation procedure, did not show a significant advantage with the current choice of embedding models and conditioning.

## 4 RELATED WORK

The primary method for evaluating text-to-image models involves gathering human judgments on a specific benchmark (i.e., a set of prompts). Previous research highlights that the composition of this benchmark significantly influences the resulting model rankings. This has led to the development of benchmarks with broader skill coverage, e.g., text rendering and spatial reasoning (Cho et al., 2023; Li et al., 2024; Wiles et al., 2024), as well as benchmarks targeting specific skills like numerical reasoning

(a) Inception embeddings.

(b) PALI(emb1) embeddings - conditioned on attribute.

(c) PALI(emb1) embeddings - conditioned on object and attribute.

Figure 8: **Ranking by autoevaluation.** Model comparisons with the Vendi Score based on (a)Inception, (b)PALI(emb1) conditioned on the attribute, and (c)PALI(emb1) conditioned on object and attribute. Each entry represents a comparison between two models. The sign indicates the model in the row is better ($>$), worse ($<$), or not significantly different ($=$) than the model in the column.

(Kajić et al., 2024). Although human evaluation remains the gold standard, numerous automatic metrics have been proposed to potentially replace human judgments, at least for certain applications (e.g., Hessel et al., 2021; Wiles et al., 2024; Huang et al., 2023; Lin et al., 2024; Senthilkumar et al., 2024). Rigorous validation of these metrics is crucial across diverse conditions, including different prompt sets, human evaluation templates, and models (Wiles et al., 2024). An important facet of evaluating text-to-image models involves measuring the diversity of their output (Dombrowski et al., 2024; Vice et al., 2024). This has resulted in different metrics, both reference-based (Sajjadi et al., 2018; Heusel et al., 2017; Salimans et al., 2016) and reference-free (Friedman & Dieng, 2022; Rassin et al., 2024; Mironov & Prokhorenkova, 2024; Ospanov et al., 2025; Limbeck et al., 2024). The advantage of reference-free metrics is their independence from a ground-truth set, which permits the evaluation of diversity in broader contexts. One such recent metric, the Vendi score (Friedman & Dieng, 2022), has influenced subsequent research (Kannen et al., 2024a; Hemmat et al., 2024; Jalali et al., 2024). Despite these developments, none of the proposed metrics have undergone thorough evaluation, frequently being tested only on generic prompts or in simplified settings. Moreover, surprisingly, the majority of previous studies lack human evaluation to demonstrate the validity of these metrics. To address this gap, we introduce a prompt set designed for evaluating diversity across particular attributes and propose and validate a human evaluation template to gather ground-truth diversity judgments. Finally, we compare existing metrics and models under various conditions.

## 5 DISCUSSION

Ensuring diversity in text-to-image (T2I) model outputs is essential, serving as a measure of their ability to express real-world variety. However, rigorous evaluation of this diversity, particularly for specific attributes, remains challenging. This paper introduces a novel framework for attribute-specific T2I diversity evaluation. It comprises a systematic prompt set and a human evaluation template, which has been validated to significantly improve the accuracy of human judgments by explicitly defining the attribute of interest. This framework provides a crucial ground truth for understanding and measuring diversity beyond general impressions. Applying this framework, we ranked prominent T2I models based on their attribute-specific diversity, identifying Imagen 3 and Flux 1.1 as strong performers. Furthermore, we leveraged our human data to evaluate automated evaluation approaches based on the Vendi Score. Our results demonstrate that the choice of embedding space, upon which autoevaluation metrics operate, is crucial for achieving results that broadly align with human judgments. Notably, our findings indicate that Vendi Score-based autoevaluation approaches can capture human-perceived diversity with approximately 80% accuracy and correctly yield similar results for pairwise model comparisons when a comparable statistical analysis methodology is employed. The broad impact of this work lies in its potential to improve T2I model quality in terms of diversity by providing an evaluation framework grounded in human perception. Moreover, unlike the previous work that often relies on attribute classifiers (e.g., gender), our evaluation methodology can be employed to measure demographic diversity in a classification-free manner in future research.

## 6 ETHICS STATEMENT

This work involved data collection from human annotators. Each one of the 20 different participants has been compensated for the time invested in the experiment according to the minimum wage in their geographical location. Before completing the annotation task, annotators were given a comprehensive set of instructions and could take as much time as necessary to complete the task.

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

APPENDIX

## A  HUMAN EVALUATION TASK DETAILS

### A.1  INSTRUCTIONS

Before completing the annotation task, annotators were given a comprehensive set of instructions including the following guidelines:

- The goal of the task is to compare the how diverse two sets of images are with respect to a given attribute;
- For the given two sets of images, answer the question about how diverse the concept is with respect to the specific attribute highlighted in the prompt;
- You should count how many different instances of a particular attribute they observe on the left and right sets of images, separately;
- For example, if the attribute is "background" and the prompt is "animal", raters should count how many different backgrounds appear in each set of images and finally judge how diversity of the two sets compares to each other with respect to this attribute;
- Finally, based on the counts, pick one of the following options: (1) Left is more diverse; (2) Right is more diverse; (3) Equally diverse; (4) Unable to answer.

Along with the written instructions, annotators were also given examples corresponding to options 1, 2, and 3.

### A.2  ADDITIONAL INFORMATION

In total, 24591 annotations were collected in our study, including the pilot runs. The average time to complete the task with the final template was 32 seconds.

## B  HUMAN EVALUATION TEMPLATE

### B.1  GOLDEN SET CONCEPT-ATTRIBUTE PAIRS

We considered the following categories and aspects of variation for the golden set: <color, flower>, <material, container>, <color, language), <background, animal>, <material, chair>, <side dish, cookie shape>, <pattern, clothing>, <style, building>, <weather, biome>, <color, vehicle>. We validate the evaluation template by comparing cases where (i) the concept remains constant across images in the set while the aspect varies: images of the same flower (rose) in all considered colors (8 images per concept); (ii) the concept varies across images while the aspect remains the same: images of all considered flowers types in the red color (8 images per concept); and (iii) both the concept and the aspect vary across images within the set: images of all flowers, each one in one of the different colors (8 images per concept). For each concept we then generate 24 different images, yielding a total of 240 images for the full golden set. In the table below we present all considered concepts and aspects of variations values. The specific values for each concept and attribute are presented in Table 1.

For each case, images were generated using Imagen 3 with the following prompt: A photorealistic image of a aspect of variation value concept value. For example, "A photorealistic image of a yellow begonia". As images were synthetically generated following a carefully crafted protocol, we could compare the performance of human annotators as well as autoraters based on multimodal language models such as Gemini in the task of evaluating for specific aspects of variation.

### B.2  USER INTERFACE SCREENSHOTS

| Concept | Concept values | Aspect of variation | Aspect of variation values |
|---|---|---|---|
| Flower | Begonia, Carnation Geranium, Hibiscus Lily, Poppy Rose, Tulip | Color | Yellow, light purple white, blue green, orange red, purple |
| Container | Beer, champagne cognac, cup doublewalled, mug shot glass, water | Material | Porcelain, metal stainless steal, ceramic glass, gold copper, plastic |
| Neon sign language | Bonjour, hello hei, oi sawubona, hola buna, ciao | Color | Blue, green orange, pink purple, red white, yellow |
| Animal | Capybara, monkey dog, snake cat, lion tree, elephant | Background | Beach, jungle park, rock room, savannah tree, water |
| Chair | Dinning, armchair office, rocking lounge, folding barstool, recliner | Material | Wood, upholstered mesh, wicker leather, metal plastic, microfiber |
| Cookie shape | Round, square crescent, start heart, diamond ghost, bat | Side dish | Milk, coffee tea, hot chocolate soda, fruits ice cream, walnuts |
| Clothing | Tshirt, dress pants, skirt jacket, gloves sweater, scarf | Pattern | Solid color blue, striped polka dot, floral plaid, checkered animal print, camouflage |
| Building | Skyscraper, residential industrial, commercial church, theater train station, school | Style | Modern, gothic victorian, art deco baroque, romanesque brutalist, traditional japanese |
| Biome | Desert, rainforest grassland, tundra swamp, coastal jungle, mountain | Weather | Sunny, cloudy rainy, snowy foggy, stormy sunset, overcast |
| Vehicle | Car, truck motorcycle, bus airplane, boat train, helicopter | Color | Red, blue green, yellow white, black orange, gray |

Table 1: Golden set generation: concepts and respective aspects of variation.

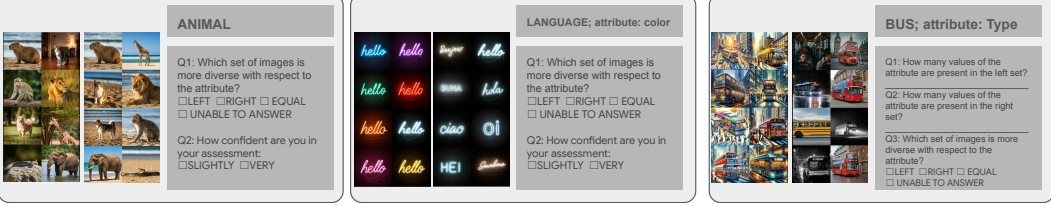

Figure 9: Examples of human evaluation templates used in the pilot study. In the template variant `w/o aspect`, only the `category` is provided. In the variant with `count`, an additional question is included for each set, prompting annotators to specify the number of distinct values observed for the target attribute within the corresponding image set. For exact examples see Figs. 10-12.

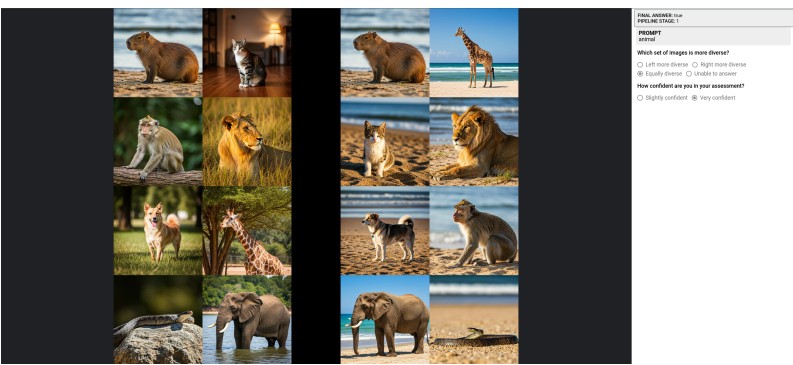

Figure 10: A screenshot of the user interface for one annotation example for the condition "No aspect".

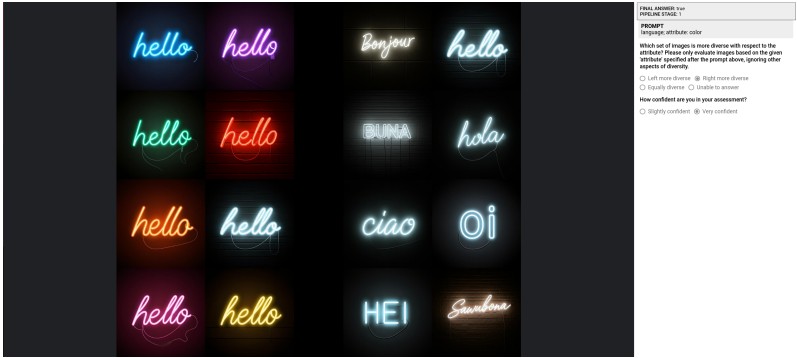

Figure 11: A screenshot of the user interface for one annotation example for the condition "Aspect".

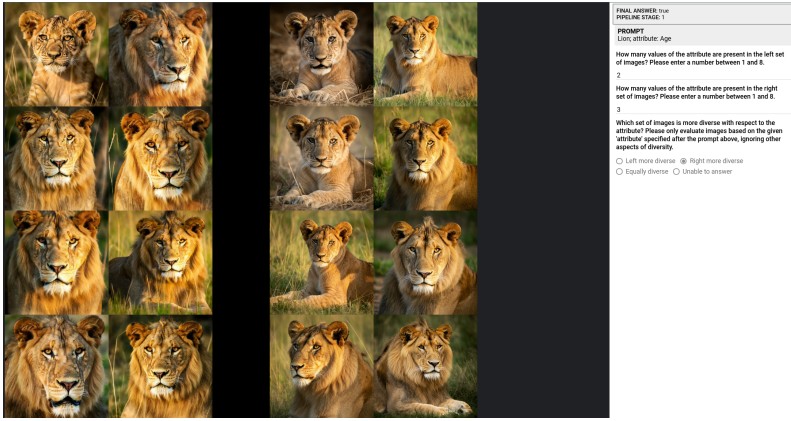

Figure 12: A screenshot of the user interface for one annotation example for the condition "Count".

## C ADDITIONAL HUMAN EVALUATION RESULTS

In Fig. 13 we show the histogram of counts averaged across the 5 raters each set in all side-by-side comparisons.

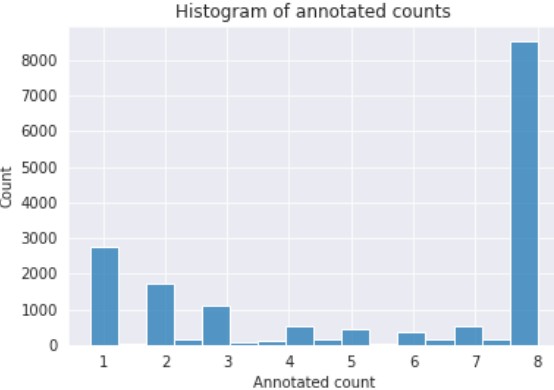

Figure 13: Distribution of all counts annotated by human raters.

### C.1 SENSITIVITY ANALYSIS ON RATER COUNT

We performed a sensitivity analysis taking into account the impact of the rater counter on $\alpha$. We subsample the annotators, obtaining sets of $K = 2, 3, 4, 5$ raters and compute $\alpha$. This process is repeated 100 times for each $K$ (except 5, the full annotators set) and report the average and 95% confidence intervals in the following table. Overall, the results show that we have high robustness in the agreement. Even with only $k = 2$ raters, the lower bound of the 95% CI rarely drops below 0.80 (the standard threshold for high reliability), and for most pairs, it stays above 0.90. The gain in mean inter-annotator agreement from $k = 3$ to $k = 5$ is marginal. This confirms that our protocol (using 5 raters) is statistically overpowered and rigorous. Moreover, the mean $\alpha$ is stable across all $k$. This indicates that our high agreement scores are not driven by a few good raters but are a consistent property of the evaluation template itself.

| Model Pair | $k = 2$ **Raters** | $k = 3$ **Raters** | $k = 4$ **Raters** | $k = 5$ **(Full)** |
|---|---|---|---|---|
| **Imagen 2.5 vs Muse 2.2** | 0.954 [0.902, 0.995] | 0.947 [0.917, 0.981] | 0.948 [0.936, 0.967] | 0.948 |
| **Imagen 2.5 vs Imagen 3** | 0.946 [0.921, 0.973] | 0.946 [0.928, 0.962] | 0.947 [0.939, 0.954] | 0.947 |
| **Imagen 2.5 vs DALLE3** | 0.968 [0.941, 0.990] | 0.969 [0.954, 0.990] | 0.969 [0.962, 0.980] | 0.969 |
| **Imagen 2.5 vs Flux 1.1** | 0.969 [0.957, 0.986] | 0.969 [0.959, 0.976] | 0.969 [0.962, 0.972] | 0.969 |
| **Muse 2.2 vs Imagen 3** | 0.956 [0.925, 0.990] | 0.954 [0.938, 0.972] | 0.954 [0.946, 0.962] | 0.954 |
| **Muse 2.2 vs DALLE3** | 0.968 [0.946, 0.998] | 0.966 [0.952, 0.995] | 0.967 [0.960, 0.980] | 0.967 |
| **Muse 2.2 vs Flux 1.1** | 0.971 [0.948, 1.000] | 0.971 [0.953, 0.989] | 0.971 [0.964, 0.978] | 0.971 |
| **Imagen 3 vs DALLE3** | 0.826 [0.780, 0.887] | 0.824 [0.794, 0.851] | 0.825 [0.810, 0.836] | 0.826 |
| **Imagen 3 vs Flux 1.1** | 0.870 [0.844, 0.890] | 0.867 [0.853, 0.883] | 0.868 [0.862, 0.877] | 0.867 |
| **DALLE3 vs Flux 1.1** | 0.915 [0.869, 0.944] | 0.912 [0.887, 0.931] | 0.911 [0.901, 0.925] | 0.911 |

Table 2: Sensitivity analysis of inter-annotator agreement ($\alpha$) with varying number of raters ($k$). We report the mean $\alpha$ and the 95% confidence interval (bootstrapped over 100 iterations for subsampled sets).

### C.2 RANKING STABILITY

We performed a bootstrap analysis by resampling concepts with replacement 1000 times and found that regardless of whether we include or not the ties, all the rankings are stable (i.e. the confidence interval does not cross 0) for the 95% confidence interval.

Table 3: Bootstrap stability analysis of model rankings (1000 resamples). We report the difference in win rates ($\Delta_{\text{winrate}}$) and 95% Confidence Intervals (CI). A ranking is considered **Stable** if the CI does not cross zero.

| Model Pair | With Ties | | | No Ties | | |
|---|---|---|---|---|---|---|
| | $\Delta_{\text{winrate}}$ | CI | Stable | $\Delta_{\text{winrate}}$ | CI | Stable |
| **Imagen 2.5 vs Muse 2.2** | $-0.056085$ | $[-0.10, -0.02]$ | True | $-0.165975$ | $[-0.28, -0.05]$ | True |
| **Imagen 2.5 vs Imagen 3** | $-0.261750$ | $[-0.31, -0.22]$ | True | $-0.563230$ | $[-0.65, -0.48]$ | True |
| **Imagen 2.5 vs DALLE3** | $-0.200514$ | $[-0.24, -0.16]$ | True | $-0.583720$ | $[-0.68, -0.49]$ | True |
| **Imagen 2.5 vs Flux 1.1** | $-0.270049$ | $[-0.31, -0.23]$ | True | $-0.716413$ | $[-0.79, -0.64]$ | True |
| **Muse 2.2 vs Imagen 3** | $-0.194932$ | $[-0.24, -0.15]$ | True | $-0.392261$ | $[-0.48, -0.30]$ | True |
| **Muse 2.2 vs DALLE3** | $-0.115269$ | $[-0.16, -0.08]$ | True | $-0.324438$ | $[-0.43, -0.22]$ | True |
| **Muse 2.2 vs Flux 1.1** | $-0.222187$ | $[-0.26, -0.18]$ | True | $-0.608733$ | $[-0.69, -0.52]$ | True |
| **Imagen 3 vs DALLE3** | $0.095448$ | $[0.05, 0.14]$ | True | $0.213689$ | $[0.12, 0.31]$ | True |
| **Imagen 3 vs Flux 1.1** | $-0.088937$ | $[-0.13, -0.04]$ | True | $-0.201318$ | $[-0.30, -0.10]$ | True |
| **DALLE3 vs Flux 1.1** | $-0.141041$ | $[-0.18, -0.11]$ | True | $-0.454561$ | $[-0.56, -0.35]$ | True |

## D  A DEEP DIVE ON OUR CURATED PROMPT SET GENERATION DETAILS

### D.1  PROMPT SET GENERATION

We used the following prompt to generate the concept-factor pairs:

Your task is to generate a dataset with prompts for evaluating text-to-image models. These prompts will be used to generate realistic images and assess the diversity of the corresponding generative model with respect to a specific aspect. All prompts should correspond to realistic images. Write on the side the main object of the prompt and the aspect diversity will be measured with respect to. Here are a few examples:

Apple. An image of an apple. Color.
Book. A photograph of a book. Thickness.
Bowl of soup. An image of a bowl of soup. Ingredients.
Bridge. A photograph of a bridge. Shape.
Building. An image of a building. Style.
Cake. A photograph of a cake. Flavour.
Car. A photograph of a car. Type.

Omit any other text.
Generate at least 95 cases.
Do not include categories that involve people.

### D.2  ON THE SUFFICIENCY OF THE PROMPT SET FOR DISCRIMINATING MODELS

In order to further show that our results are significant with the current set, we ran new versions of the model comparison with the human annotations presented in Sec. 3 with versions of our prompt set that have a smaller number of concepts.

More specifically, we repeated the Binomial tests (at the same significance level) after randomly removing an increasing amount of concepts, which resulted in prompt sets of size 74, 64, 54, and 24 concepts. Overall, we find that decreasing the prompt set size to 74 concepts doesn't affect any of the results. As the prompt set size further decreases, we start to see the results changing as the number of significant pairwise comparisons decreases. We observe that drastically decreasing the prompt set size makes the data no longer able to capture significant differences between models such as Imagen 3 and Imagen 2.5, as expected.

In the Table 4, we show the results of the Binomial tests for the 5 different sizes of prompt set, including the full set, from left to right (i.e. the first symbol represents the result with the full set as in

Fig. 5b, the second symbol the result with 74, then 64, 54, and 24 concepts). Notice that even with the smallest set we don't see a contradiction in the ordering.

|  | Flux 1.1 | Imagen 3 | DALLE3 | Muse 2.2 | Imagen 2.5 |
|---|---|---|---|---|---|
| Flux 1.1 | — | =, =, >, =, < | >, >, =, =, = | >, >, >, >, > | >, >, >, =, > |
| Imagen 3 |  | — | =, =, <, =, > | =, =, =, =, = | >, >, >, >, = |
| DALLE3 |  |  | — | <, <, =, =, = | =, =, =, =, = |
| Muse 2.2 |  |  |  | — | <, <, <, <, < |
| Imagen 2.5 |  |  |  |  | — |

Table 4: Repeating model comparisons across smaller versions of our prompt set. Decreasing the prompt set size to 74 concepts doesn't affect any of the results.

# E  ADDITIONAL AUTOEVALUATION RESULTS

## E.1  COMPUTE USAGE

We used accelerators for running automatic evaluation metrics and generating the images. We run all metrics on a TPU V3 hardware[2]. The image generation pipeline ran on 4 TPUs.

## E.2  PERFORMANCE FOR DETECTING EQUALLY DIVERSE IMAGE SETS

We evaluate how good embeddings are at detecting equally diverse image sets. To not have a threshold-dependent metric, we use the area-under-the-ROC curve (AUC). We construct the true binary label as whether the image sets are labelled as equally diverse or not. We construct the scores as the absolute difference between the metric scores. We then plot the AUC. A good metric would have an AUC close to one, indicating that when the differences are small, the image sets are more likely to have been labelled as the same by the human annotators. We plot results in Figure 14, and find that no metric performs particularly well (AUC < 0.6 in all cases). However, the IMAGENET INCEPTION one performs best, presumably as it is trained to be invariant to small differences and so, as we can see in Figures 15-16, as a lack of diversity usually arises when images are very similar, the embedding performs well. However, we hypothesise that in the face of confounders (e.g. we want to measure diversity of the color of an object but not the type of object), we would not expect such an embedding to do well.

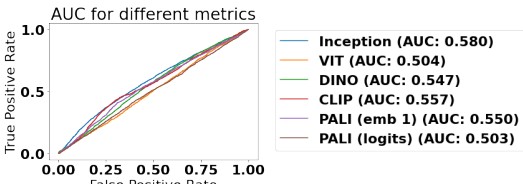

Figure 14: AUC to measure metrics ability to identify sets of equal diversity. It is clear that no metric is particularly effective at differentiating visually similar versus not sets of images.

## E.3  ADDITIONAL QUALITATIVE RESULTS

In Figs. 15 and 16 we visualize examples for four side-by-side comparisons where the corresponding autoraters indicate that a group of images have highest or lowest diversity. We can see that results are reasonable and that in general, images with low diversity arise due to mode collapse, i.e. the model generates a very similar image for the same concept. This could explain why the INCEPTION model performs poorly on the pilot data but well on the model comparison data. INCEPTION features are effective for identifying these issues but no effective for identifying diversity in the case of confounding aspects (e.g., the background is changing while the animal is staying the same).

---

[2]https://cloud.google.com/tpu/docs/v3

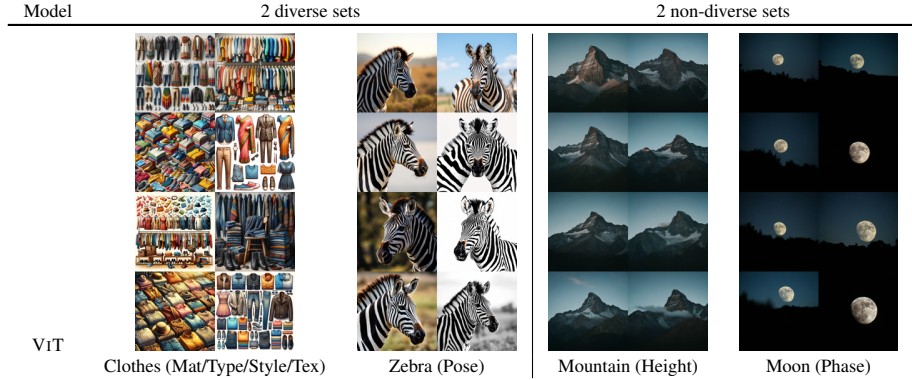

Figure 15: Qualitative results for different autoraters on the T2I annotated dataset, showing two very diverse and two non diverse sets as determined by the ViT-based autorater.

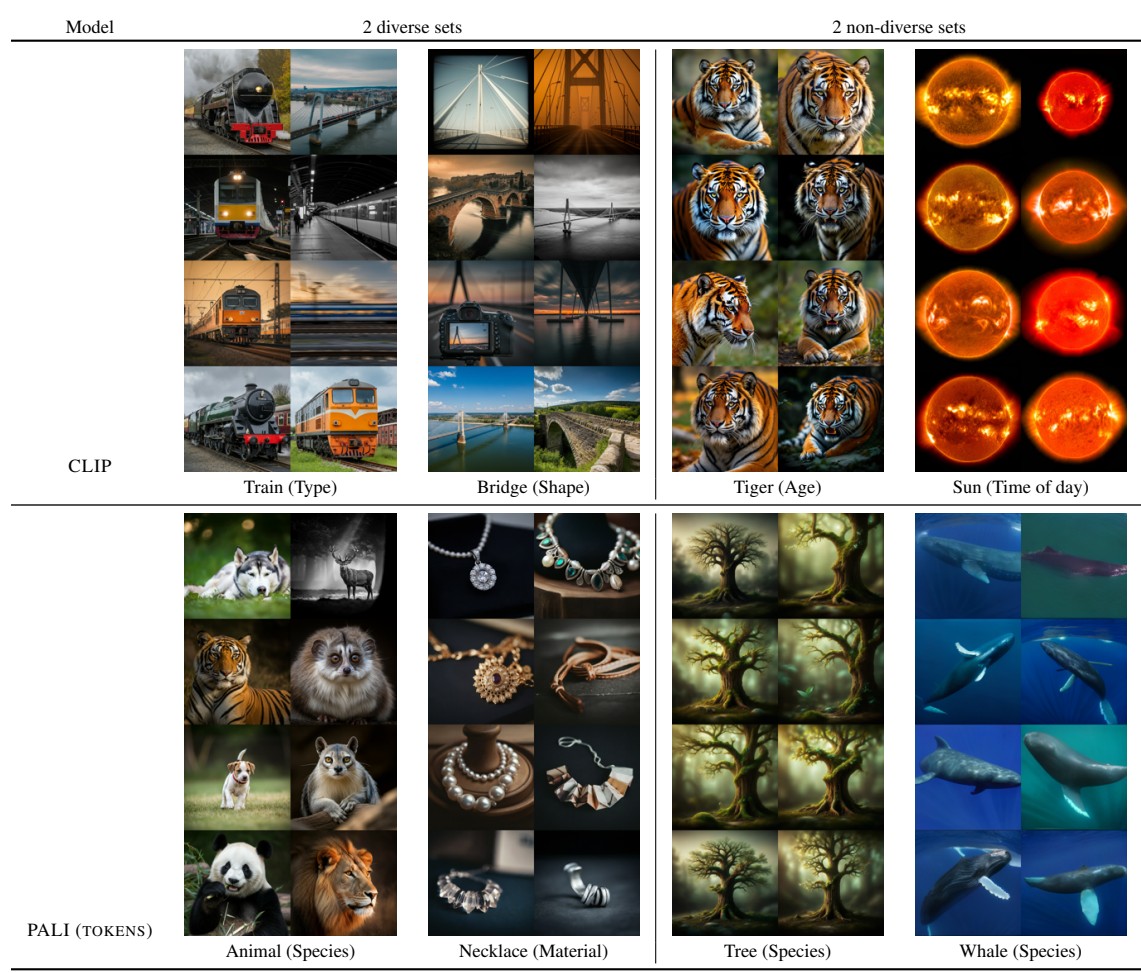

Figure 16: Qualitative results for different models, showing two very diverse and two non diverse sets.

### E.4 IMPACT OF THE PROMPT FOR THE MULTIMODAL EMBEDDINGS

We explore how the choice of prompt impacts results for the multimodal embeddings. We explore four different prompts which differ in their specificity and relatedness to the attributes under question. [attribute] and [object] are placeholders and filled in based on the object / attribute under test. The templates we consider are as follows:

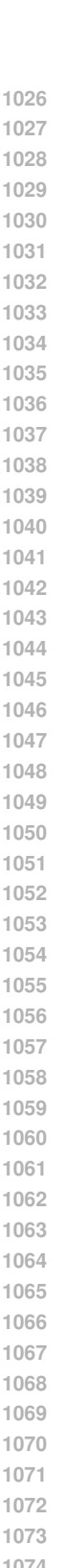
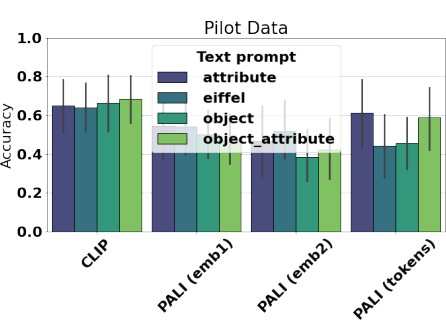
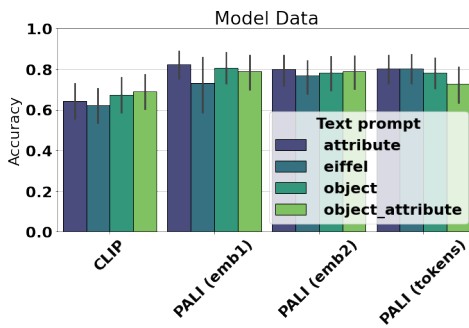

(a) Results on the "diverse" golden set.

(b) Results on the annotation set, where annotators see count differences > 4.

Figure 17: Additional auto-eval results that show how results vary based on the textual prompt for the multimodal embeddings. We can see that we *do not* see consistently better results with more related prompts (`What is the [attribute] of the [object]?`, `What is the [attribute]?`), implying the textual input is being ignored.

1. OBJECT_ATTRIBUTE: `What is the [attribute] of the [object]?`
2. ATTRIBUTE: `What is the [attribute]?`
3. OBJECT: `What is the [object]?`
4. EIFFEL: `Where is the Eiffel Tower?`

We would expect the first two questions to be most effective as they directly ask about the property for which we are measuring diversity. The object may be related but can be a confounder and the "Eiffel Tower" question is unrelated.

Results are shown in Figure 17. Surprisingly, we find that we do not see consistent benefit from the two most related prompts (OBJECT_ATTRIBUTE, ATTRIBUTE), implying that the embeddings are mostly vision based. A more controllable multimodal embedding we hypothesise would be more effective in this setting.

### E.5 MODEL RANKING WITH AUTOEVALUATION APPROACHES

In this section, we include more results for model ranking based on our auto-evaluation approaches:

- Figures 18, 19 and 20 show the results of compare model rankings in terms of significance in the number of wins with Wilcoxon signed-rank tests under a 95% confidence level using additional models to compute embeddings. This figure completes Figure 8 in Sec. 3.4. In theses figures, we can see:
  - Model ranking based on other embeddings. We observe that similarly to the observations in Sec. 3.4, for all embeddings except IMAGENET VIT, Imagen3 is not worse than all other models. We also observe that independently of the choice of embedding, Flux1.1, Imagen3 and DALLE3 are not worse than Muse2.2 and Imagen2.5. The differences between the models in the top group and the bottom group are more or less detected depending on the embeddings.
  - As mentioned in the main text, we also see the differences between multimodal models. These results highlight how the influence of the choice of embedding models and of conditioning on the model ranking results.
- Figures 21, 22 and 23 show the win rates corresponding to the results shown in Figure 8 in Sec. 3.4 and the additional results described above on the left panels, and compare the distributions of the two best and closest models in terms of behavior according to human evaluation, Imagen3 and Flux1.1, on the right panels. These figures correspond respectively to image models, multimodal model conditioned on attributes, and multimodal models conditioned on objects and attributes.

**(a) ViT embeddings.**

| | Flux 1.1 | Imagen 3 | DALLE3 | Muse 2.2 | Imagen 2.5 |
|---|---|---|---|---|---|
| Flux 1.1 | × | > | = | > | > |
| Imagen 3 | < | × | < | = | > |
| DALLE3 | = | > | × | > | > |
| Muse 2.2 | < | = | < | × | > |
| Imagen 2.5 | < | < | < | < | × |

**(b) DINO embeddings.**

| | Flux 1.1 | Imagen 3 | DALLE3 | Muse 2.2 | Imagen 2.5 |
|---|---|---|---|---|---|
| Flux 1.1 | × | < | > | > | > |
| Imagen 3 | > | × | > | > | > |
| DALLE3 | < | < | × | > | > |
| Muse 2.2 | < | < | < | × | > |
| Imagen 2.5 | < | < | < | < | × |

Figure 18: **Model ranking using auto evaluation approaches with additional image models.** We compare model rankings in terms of significance in the number of wins with Wilcoxon signed-rank tests under a 95% confidence level. Each entry in the each of the grids represents a comparison between two models. The > sign indicates the model in the row is better, worse (<), or not significantly different (=) than the model in the column. The win rates in each of the grids are computed using the scores based on (a) IMAGENET VIT embeddings and (b) DINO embeddings.

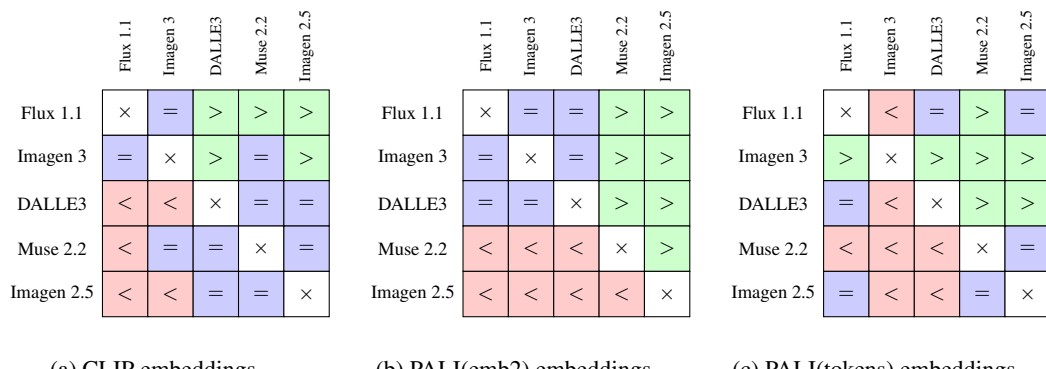

**(a) CLIP embeddings.**

| | Flux 1.1 | Imagen 3 | DALLE3 | Muse 2.2 | Imagen 2.5 |
|---|---|---|---|---|---|
| Flux 1.1 | × | = | > | > | > |
| Imagen 3 | = | × | > | = | > |
| DALLE3 | < | < | × | = | = |
| Muse 2.2 | < | = | = | × | = |
| Imagen 2.5 | < | < | = | = | × |

**(b) PALI(emb2) embeddings.**

| | Flux 1.1 | Imagen 3 | DALLE3 | Muse 2.2 | Imagen 2.5 |
|---|---|---|---|---|---|
| Flux 1.1 | × | = | = | > | > |
| Imagen 3 | = | × | = | > | > |
| DALLE3 | = | = | × | > | > |
| Muse 2.2 | < | < | < | × | > |
| Imagen 2.5 | < | < | < | < | × |

**(c) PALI(tokens) embeddings.**

| | Flux 1.1 | Imagen 3 | DALLE3 | Muse 2.2 | Imagen 2.5 |
|---|---|---|---|---|---|
| Flux 1.1 | × | < | = | > | = |
| Imagen 3 | > | × | > | > | > |
| DALLE3 | = | < | × | > | > |
| Muse 2.2 | < | < | < | × | = |
| Imagen 2.5 | = | < | < | = | × |

Figure 19: **Model ranking using auto evaluation approaches with additional vision and language models conditioned on attributes.** We compare model rankings in terms of significance in the number of wins with Wilcoxon signed-rank tests under a 95% confidence level. Each entry in the each of the grids represents a comparison between two models. The > sign indicates the model in the row is better, worse (<), or not significantly different (=) than the model in the column. The win rates in each of the grids are computed using the scores based on (a) CLIP embeddings, (b) PALI(emb2) embeddings, and (c) PALI(tokens) embeddings. All models are conditioned on attributes.

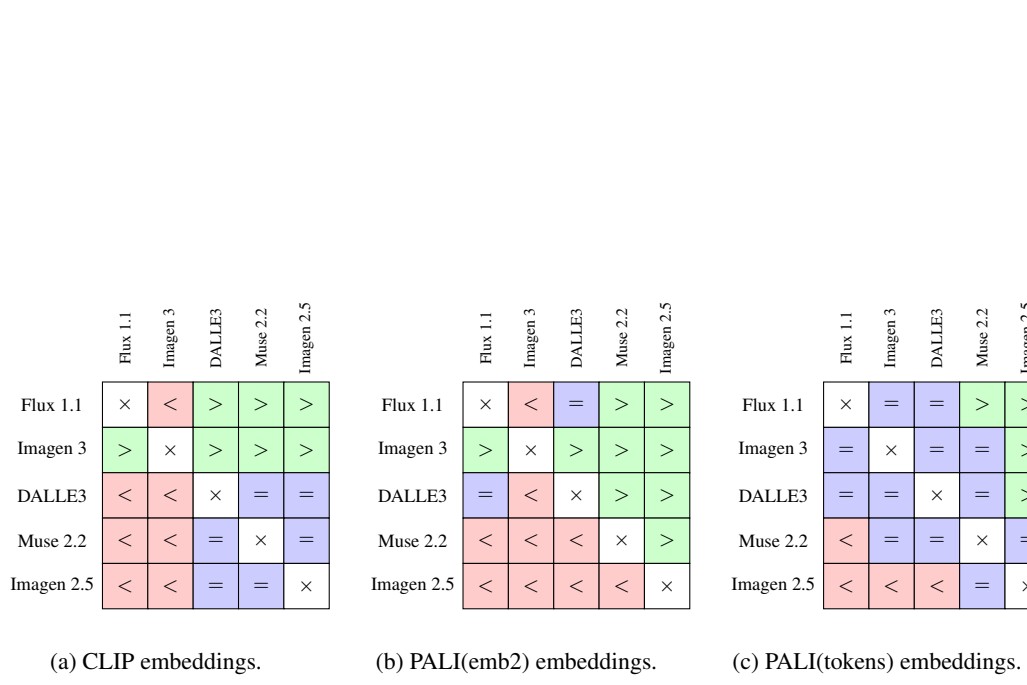

(a) CLIP embeddings.  (b) PALI(emb2) embeddings.  (c) PALI(tokens) embeddings.

Figure 20: **Model ranking using auto evaluation approaches with additional vision and language models conditioned on objects and attributes.** We compare model rankings in terms of significance in the number of wins with Wilcoxon signed-rank tests under a 95% confidence level. Each entry in the each of the grids represents a comparison between two models. The $>$ sign indicates the model in the row is better, worse ($<$), or not significantly different ($=$) than the model in the column. The win rates in each of the grids are computed using the scores based on (a) CLIP embeddings, (b) PALI(emb2) embeddings, and (c) PALI(tokens) embeddings. All models are conditioned on objects and attributes.

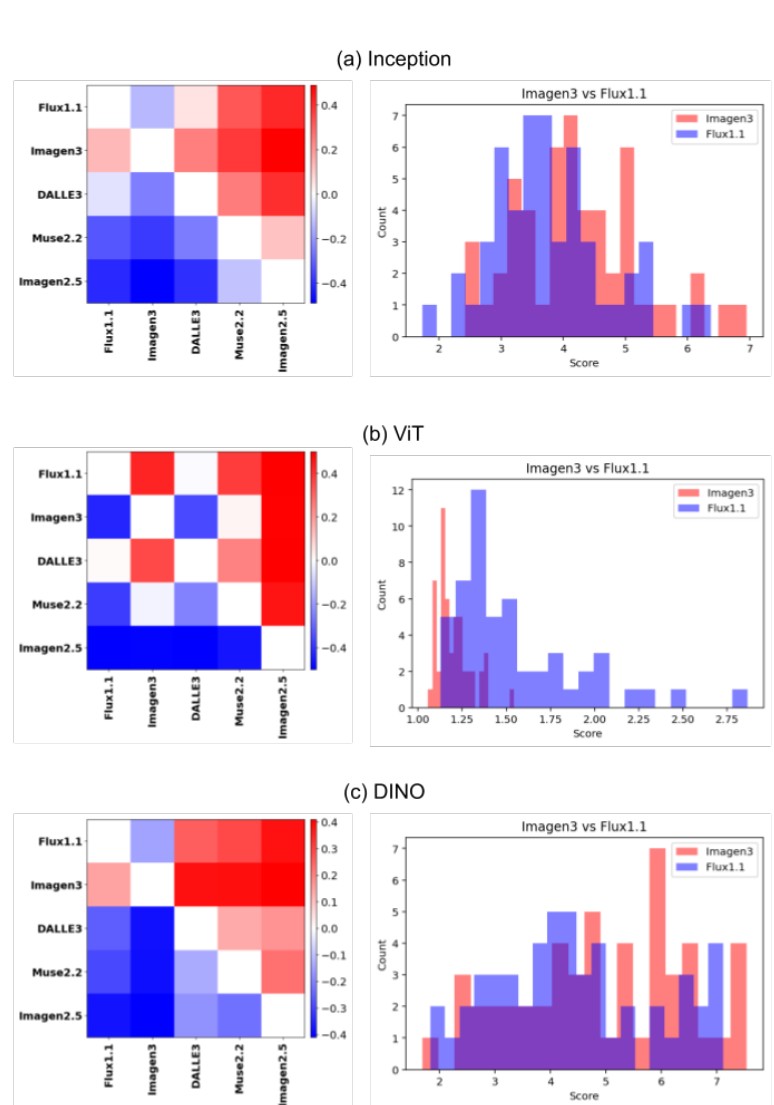

Figure 21: **Model ranking using auto evaluation approaches.** Win rate matrices and score distributions for Flux1.1 and Imagen3 using image models to compute embeddings.

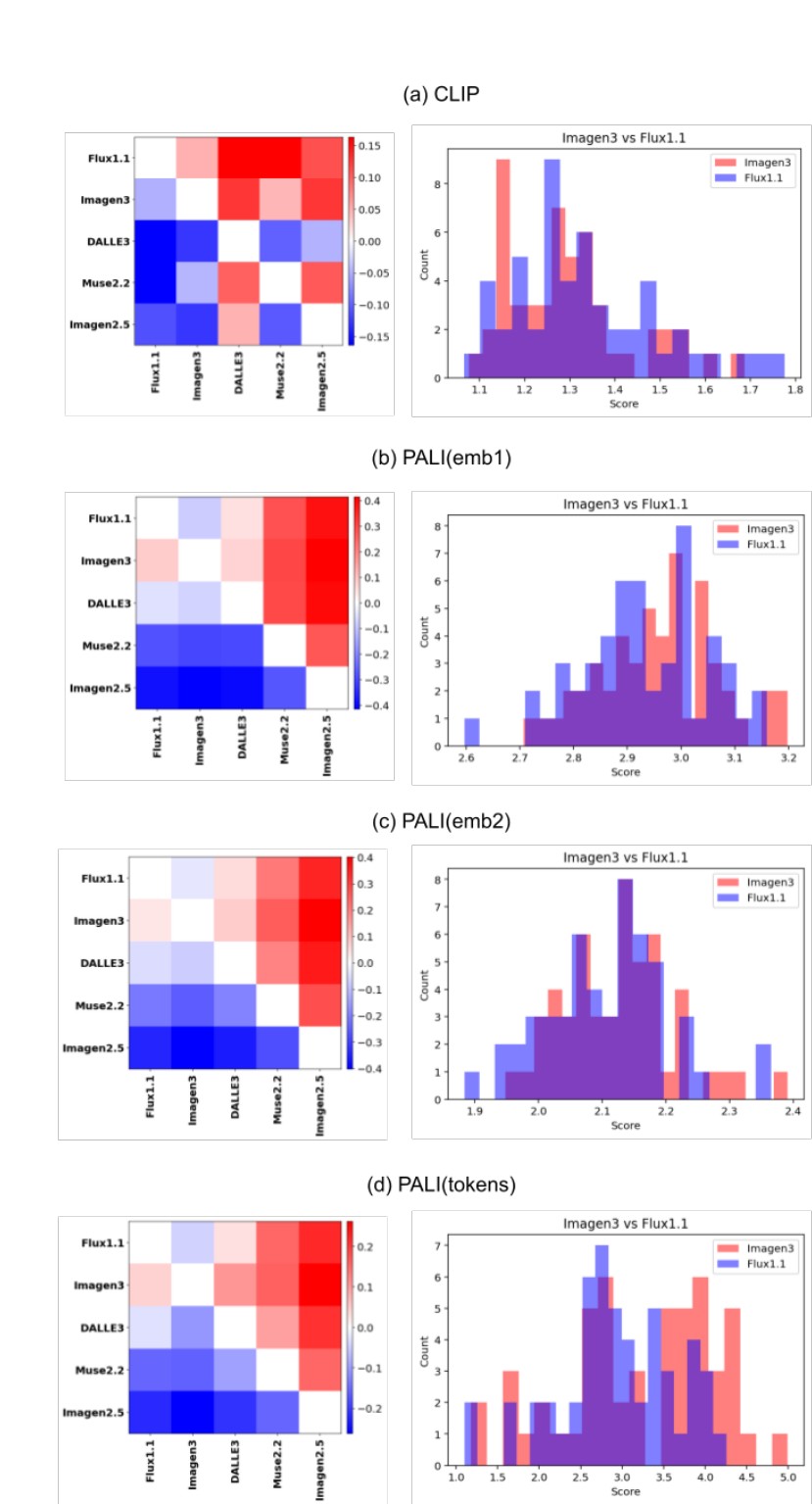

Figure 22: **Model ranking using auto evaluation approaches.** Win rate matrices and score distributions for Flux1.1 and Imagen3 using text-conditioned multimodal models to compute embeddings, conditioned on attributes.

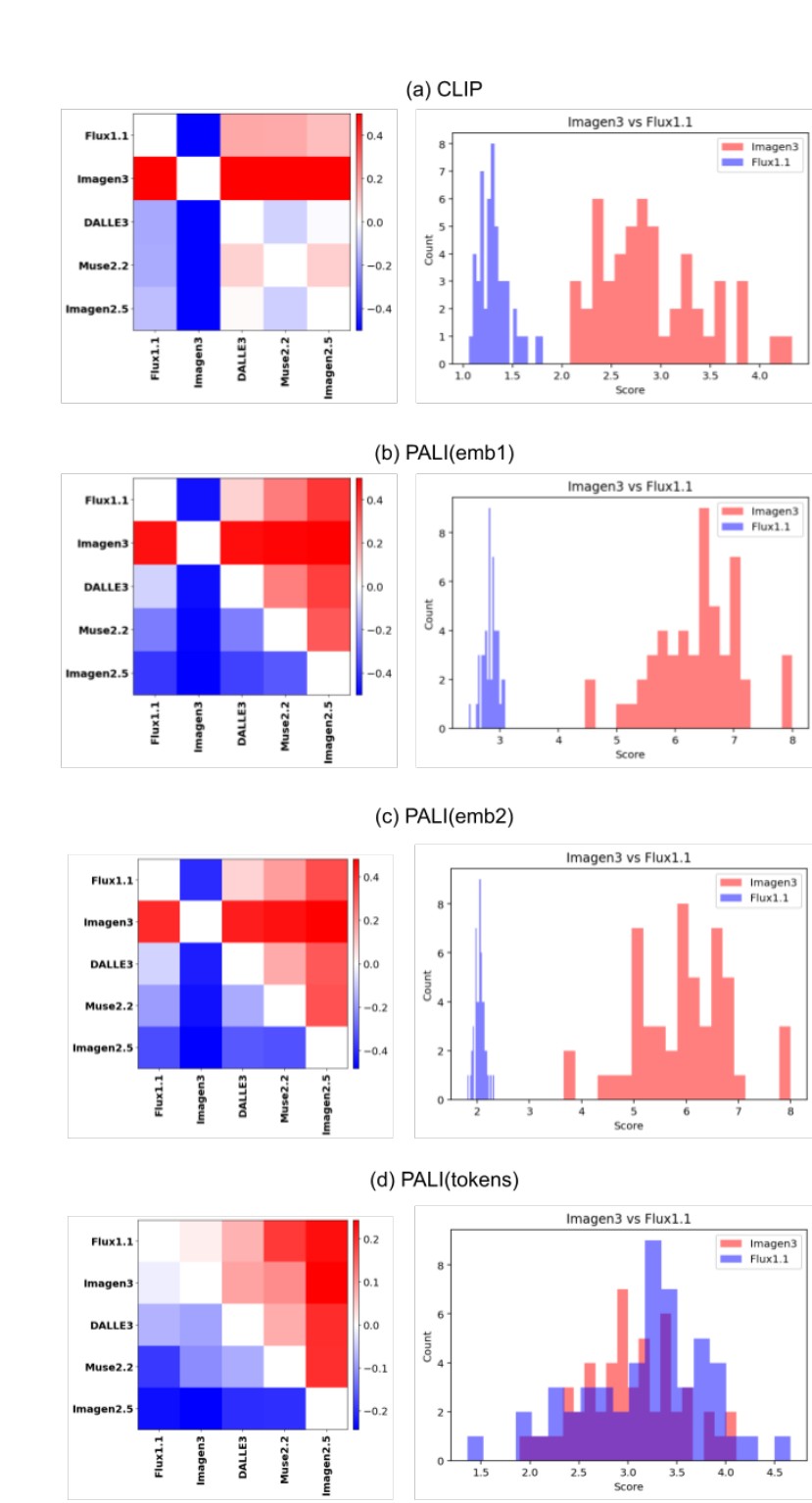

Figure 23: **Model ranking using auto evaluation approaches.** Win rate matrices and score distributions for Flux1.1 and Imagen3 using text-conditioned multimodal models to compute embeddings, conditioned on objects and attributes.

### E.6 EVALUATING DIVERSITY USING FOUNDATION MODELS

We use the following instruction: "*I am currently comparing two models with the prompt [prompt] and I would like to know which model generates more diverse images with respect to the attribute [attribute], while disregarding any other attribute in the images. In the following image I show [number of images] images generated by one model in the left, which is [model in the left side] and [number of images] images generated by another model in the right, which is [model in the right side]. You must count the number of different instances of [attribute] in both sets and use this information to decide which set is the most diverse. If there is a set of images which is more diverse than the other with respect to [attribute], can you tell me which one is the most diverse set and explain why? Any other aspects in the images besides [attribute] must not be taken into account. You can also respond that both sets are equally diverse.*."

In addition to the instruction, similarly to the human evaluation, two sets of images are given to the model as input.

## F  ABSENCE OF A DIVERSITY-FIDELITY TRADE-OFF

Evaluating the diversity of generative models presents a unique challenge: a model can trivially achieve high diversity by producing random noise–the generated noisy images are always different in a high dimensional space. Therefore, any meaningful assessment of diversity must be predicated on the assumption that the models in question are capable of generating images of sufficient quality. This quality criterion implies that the generated images must not only be visually coherent and free from significant artifacts but also effectively capture the salient aspects and core intent of the given prompt. Without this foundational understanding of quality and adherence to prompt specifications, a high diversity score would be misleading, indicating a lack of control and semantic understanding rather than a beneficial range of outputs. To illustrate this for some of the strong models we considered in our work, we compute the state-of-the-art text-to-image alignment metric Gecko (Wiles et al., 2024) for the same images used in our study in Table 5. Results show that models achieve the same average Gecko score (higher is better, 1 is the maximum) indicating they not only have strong performance in terms of text-to-image alignment, but are not statistically different in terms of this evaluation aspect. Notably, our diversity evaluation in Sec. 3 and  E showed that Imagen 3 is significantly better than both Imagen 2.5 and Muse 2.2.

| Model | Gecko | 95% CI lowerbound | 95% CI upperbound |
|---|---|---|---|
| Muse 2.2 | 0.9591 | 0.9530 | 0.9646 |
| Imagen 3 | 0.9591 | 0.9527 | 0.9647 |
| Imagen 2.5 | 0.9591 | 0.9527 | 0.9645 |

Table 5: Alignment results for models with different diversity.

## G  LLM USE DISCLOSURE

An LLM was used for polish writing of some paragraphs of the manuscript and improving the phrasing of a few sentences. No LLM was used to write extended parts of the paper, or for writing sentences from scratch, retrieval, discovery and research ideation.

