# OpenReview forum: "Benchmarking Diversity in Image Generation via Attribute-Conditional Human Evaluation"
_ICLR.cc/2026/Conference — Submitted to ICLR 2026_

### Official Review · Reviewer_8oVG · 2025-10-31

**Soundness:** 2
**Presentation:** 3
**Contribution:** 2
**Rating:** 4
**Confidence:** 4

**Summary:**

This paper addresses the challenge of rigorously evaluating diversity in text-to-image generative models by proposing an attribute-conditional human evaluation framework. The authors argue that existing diversity assessments are ambiguous and often conflate fidelity and diversity, leading to unreliable conclusions when variation factors are unspecified. To fix this, they formalize diversity along specific concepts and their key factors of variation (e.g., color of apples), develop a systematically generated prompt set spanning 86 concept-attribute pairs, and design a validated human evaluation template that improves annotation accuracy by explicitly focusing raters on a defined attribute. Using this benchmark, they collect 24,591 human annotations comparing five leading T2I models and show statistically significant model ranking differences—identifying Imagen 3 and Flux 1.1 as the most diverse models. They also evaluate the Vendi Score with various embedding choices, finding that ~65–80% alignment with humans is possible depending on representation space, while revealing sensitivity to embedding selection. Overall, the work provides a principled, scalable method for measuring and comparing model diversity and establishes a valuable dataset for validating future automated metrics.

**Strengths:**

The paper convincingly shows that diversity evaluation is ambiguous without specifying both the concept and the factor of variation.

A systematically generated prompt set covering 86 concept-attribute pairs and a large-scale human-rated dataset (24,591 comparisons), filling an evaluation gap in T2I research.

A well-designed human evaluation template that significantly improves annotation accuracy by focusing raters on explicit attributes.

**Weaknesses:**

Real-world diversity often involves interactions between multiple attributes (e.g., shape + color + background). The framework isolates each property and does not consider compositional or multi-attribute diversity, leaving a large portion of the diversity landscape unexplored.

The benchmark focuses on everyday categories (food, nature, manufactured objects), often static and object-centric. This might underrepresent scenarios like human attributes, fine-grained scenes, and spatial relations, and overfit evaluation to objects common in existing datasets (ImageNet bias). As a result, findings may not generalize to domain-specific or creative applications.

More diverse does not equal more realistic. A model generating incoherent or noisy variations could superficially score high, and the framework does not explicitly guard against trade-offs between diversity and quality.

**Questions:**

How does the framework generalize to multi-attribute or compositional variation?

What is the framework’s sensitivity to prompt rephrasing or category expansion (e.g., uncommon or artistic concepts)?

Would incorporating vision-language grounding models trained for attribute classification improve metric-human alignment?

---

> ### Author Response · Authors · 2025-11-18
> **Response to Reviewer 8oVG (1/2)**
>
> We thank the reviewer for their assessment of our work and for recognizing the value of our systematically generated prompt set, the scale of our human evaluation (24k+ annotations), and the effectiveness of our template in reducing ambiguity. We appreciate the thoughtful critique regarding the scope of diversity and the relationship between diversity and quality. Below, we address the specific concerns and questions raised.
>
> **1. On multi-attribute and compositional Diversity (Weakness 1 & Q1):** The reviewer notes that our framework isolates properties and does not strictly measure compositional diversity. We acknowledge that real-world diversity is compositional; however, we argue that **isolating atomic attributes is the necessary first step toward rigorous measurement**.
>
> As demonstrated in Figure 1, underspecified tasks (e.g., "Which set is more diverse?") lead to inconclusive human labels because annotators implicitly prioritize different attributes. To measure compositional diversity scientifically, one must first be able to measure atomic diversity reliably. Our work provides this foundational "unit test" for generative models. By validating that models can vary along a single, controlled axis (e.g., color) without conflating it with others (e.g., shape), we establish the ground truth necessary to build complex, multi-attribute benchmarks in the future. We view our contribution as the "controlled setting" required before moving to "in-the-wild" compositional evaluation.
>
> **2. Domain coverage and "static" objects (Weakness 2):** The reviewer concerns that the focus on everyday categories (ImageNet-like) might underrepresent creative or human-centric scenarios.
>   - **Exclusion of demographic attributes:** We explicitly excluded demographic attributes and human-related categories to avoid the ethical complexities and potential biases associated with demographic diversity in a technical benchmark. However, as noted in our Discussion, the methodology is content-agnostic and can be adapted to measure demographic diversity classification-free.
>   - **Objectivity vs. creativity:** We prioritized "everyday" concepts (Table 1) because they offer objective ground truth. Evaluating the diversity of "artistic concepts" is inherently subjective. To benchmark models and metrics robustly, we require scenarios where the presence of a variation (e.g., is the apple red or green?) is indisputable, rather than subject to interpretation.
>
> **3. On the quality-diversity trade-off (Weakness 3):** The reviewer raises a critical point: "More diverse does not equal more realistic." We fully agree that diversity metrics must not reward noise or incoherence. Crucially, we address this specific trade-off in **Appendix F** ("Absence of a Diversity-Fidelity Trade-off").
> We evaluated the text-to-image alignment and fidelity of the models using the Gecko benchmark. As shown in Table 3, the models we compared (Muse 2.2, Imagen 3, Imagen 2.5) achieved statistically indistinguishable alignment scores (~0.959). Despite identical fidelity/alignment performance, their diversity scores varied significantly. This confirms that the diversity gaps we report are not a result of models trading off quality for random noise, but rather reflect genuine differences in generative coverage.

---

> > ### Author Response · Authors · 2025-11-18
> > **Response to Reviewer 8oVG (2/2)**
> >
> > **4. Specific Questions**
> >   - **Sensitivity to category expansion and prompt rephrasing:** We tested the robustness of the results to prompt set size in Appendix D.2, finding that our ranking results hold even when randomly removing up to ~30% of concepts. Additionally, we performed a new experiment with bootstrap analysis for the categories. By resampling concepts with replacement 1000 times, we found that regardless of whether we include or not the ties, all the rankings are stable (i.e. the CI doesn't cross 0) for the 95% CI.
> >
> > | Model Pair | $\Delta_{winrate}$ (With Ties) | CI (With Ties) | Stable (With Ties) | $\Delta_{winrate}$ (No Ties) | CI (No Ties) | Stable (No Ties) |
> > | :--- | :--- | :--- | :--- | :--- | :--- | :--- |
> > | **Imagen 2.5 vs Muse 2.2** | -0.056085 | [-0.10, -0.02] | True | -0.165975 | [-0.28, -0.05] | True |
> > | **Imagen 2.5 vs Imagen 3** | -0.261750 | [-0.31, -0.22] | True | -0.563230 | [-0.65, -0.48] | True |
> > | **Imagen 2.5 vs DALLE3** | -0.200514 | [-0.24, -0.16] | True | -0.583720 | [-0.68, -0.49] | True |
> > | **Imagen 2.5 vs Flux 1.1** | -0.270049 | [-0.31, -0.23] | True | -0.716413 | [-0.79, -0.64] | True |
> > | **Muse 2.2 vs Imagen 3** | -0.194932 | [-0.24, -0.15] | True | -0.392261 | [-0.48, -0.30] | True |
> > | **Muse 2.2 vs DALLE3** | -0.115269 | [-0.16, -0.08] | True | -0.324438 | [-0.43, -0.22] | True |
> > | **Muse 2.2 vs Flux 1.1** | -0.222187 | [-0.26, -0.18] | True | -0.608733 | [-0.69, -0.52] | True |
> > | **Imagen 3 vs DALLE3** | 0.095448 | [0.05, 0.14] | True | 0.213689 | [0.12, 0.31] | True |
> > | **Imagen 3 vs Flux 1.1** | -0.088937 | [-0.13, -0.04] | True | -0.201318 | [-0.30, -0.10] | True |
> > | **DALLE3 vs Flux 1.1** | -0.141041 | [-0.18, -0.11] | True | -0.454561 | [-0.56, -0.35] | True |
> >
> > In terms of prompt rephrasing, we investigated this in Appendix E.4 with respect to metric computation. We tested four different prompt templates for extracting embeddings (e.g., "What is the [attribute]?" vs. "What is the [attribute] of the [object]?"). We found that results were relatively stable and that increasing prompt specificity did not consistently improve metric alignment, suggesting that current vision-language embeddings are primarily vision-dominated.
> >   - **Would incorporating vision-language grounding models improve alignment?** We explored this in **Section 3.2** and **Appendix E**. We evaluated metrics based on PaLI (multimodal) against purely visual models (ViT, DINO). Interestingly, we found that while PaLI embeddings performed well, they did not significantly outperform optimized visual embeddings like Inception or ViT on the model data comparisons. However, in **Section E.6**, we tested using Gemini v2.5 Flash as a direct autorater (evaluating the images via prompt rather than embedding distance). This approach achieved 92.6% accuracy on the Golden Set, surpassing human annotators. This suggests that LLM/VLM-as-a-judge is a highly promising direction that our framework validates.
> >
> > We believe our work offers a necessary correction to the field's reliance on underspecified diversity metrics. By isolating variables, we provide the first rigorous "ruler" for this problem, upon which future compositional studies can be built. We are working on updating the manuscript to reflect the clarifications to the reviewer’s comments and we hope this, along with the remaining points discussed in the rebuttal, strengthens your support for the paper.

---

### Official Review · Reviewer_AAUy · 2025-10-31

**Soundness:** 3
**Presentation:** 3
**Contribution:** 2
**Rating:** 4
**Confidence:** 4

**Summary:**

The paper suggests that diversity in T2I should be measured relative to an explicit attribute (e.g., color, species) rather than globally. It proposes a simple human-evaluation template shows two 8-image sets for a concept, name the target attribute, has raters count distinct values, then choose Left/Right/Equal/Unable, and validates that this count-then-compare instruction improves reliability on a small golden set. A larger study across several models and many concept–attribute pairs aggregates per-concept wins with binomial tests (Imagen 3 and Flux 1.1 typically strongest). For automation, the authors reuse Vendi with different embeddings; image-only embeddings (ViT/DINO/Inception) align best with human judgments on clear-gap cases, while tie detection is weak, and text conditioning adds little.

**Strengths:**

1) Attribute-conditioned, count-anchored human evaluation reduces ambiguity and improves rater reliability; the procedure is easy to replicate.
2) Comparison shows image-only embeddings with Vendi track human judgments on large-gap cases, offering a usable baseline for fast iteration.

**Weaknesses:**

1) Limited Novelty; contributions are primarily protocol + dataset + empirical comparisons.
2) The paper does not thoroughly analyze where autoraters fail (e.g., ties and small gaps, rare attribute values, per-concept heatmaps or illustrative disagreements).

**Questions:**

1) Why were other diversity metrics or Vendi variants (e.g., quality-weighted) not considered?
2) Why are the autoraters' failure cases not probed thoroughly? If failures might be due to the data, using SigLip2 might help. In general, adding results with SigLip is better. Ideally, it would be better if a new metric were proposed after analysing the failure cases to overcome them.
3) Appendix E.6 suggests the MLLM judge (Gemini 2.5 Flash) can outperform the embedding-based Vendi setup on the golden set. Could you expand this and analyze whether, in practice, one should prefer Vendi or an MLLM judge for diversity measurement going forward?

---

> ### Author Response · Authors · 2025-11-18
> **Response to Reviewer AAUy (1/2)**
>
> We thank the reviewer for the positive assessment of our method’s soundness and presentation, and for recognizing that our count-anchored evaluation protocol effectively improves rater reliability. We appreciate the constructive feedback regarding the analysis of autorater failures and the comparison with MLLM judges. Below, we address the concerns regarding novelty and expand on the implications of our findings for the future of diversity evaluation.
>
> **1. Novelty and contribution (Weakness 1)**: The reviewer characterizes the work as "primarily protocol + dataset + empirical comparisons" and rates the contribution as "Fair." We respectfully argue that **rigorous measurement protocols are the prerequisite for scientific progress, not merely an implementation detail.**
> The current literature on generative diversity is saturated with metrics (FID, Vendi Score variants, CLIP-score distributions) that have been deployed without a reliable ground truth. By demonstrating that global diversity is an ill-posed problem that yields random human baselines, and by replacing it with a validated, attribute-conditional framework, we provide the *necessary correction* to how the field evaluates generative models. We believe establishing the "ruler" (our validated human benchmark) is a significant contribution that enables the development of better tools (automated metrics) in the future. Without this dataset, proposing a new metric is essentially guessing in the dark.
>
> **2. Failure analysis of autoraters & SigLIP (Weakness 2 & Q2)**: The reviewer asks why we did not probe failure cases more thoroughly or propose a new metric to overcome them, and suggests SigLIP.
>   - **Why embeddings fail:** We did analyze failure modes, specifically regarding "ties" (Figure 14). The core failure of embedding-based metrics (ViT, DINO, CLIP) is *invariance to nuance*. These models are trained to map an image of a "red apple" and a "green apple" close together (classification invariance) or close to the text "apple" (semantic invariance). However, diversity requires measuring that distance. Our results rigorously quantify this limitation: current SOTA embeddings are good at detecting mode collapse (large gaps) but fail at detecting subtle attribute variation (ties/small gaps).
>   - **SigLIP:** We agree that SigLIP is a promising encoder. However, our goal was to benchmark the standard encoders currently used in diversity metrics (like CLIP and DINOv2) to establish a baseline for the community. We hypothesize SigLIP would perform similarly to PaLI/ViT—better than CLIP, but still limited by the "embedding bottleneck" discussed below.
>   - **Why no new metric?** We deliberately chose not to propose a complex new heuristic metric because our findings point to a different conclusion: the future is not embedding algebra, but VLM judges. (See point 4).
>
> **3. Why not quality-weighted Vendi Score? (Q1):** We intentionally avoided quality-weighted metrics to isolate the variable of interest: *diversity*. As noted in the paper (and validated by our Gecko scores in Table 3), the models we compared (Imagen 3, Muse 2.2, etc.) have converged to very similar (and high) fidelity scores. Quality-weighted diversity metrics tend to conflate these two orthogonal axes. If a metric scores low, is it because the model is repetitive (low diversity) or because it is blurry (low quality)? Our framework is designed to answer the diversity question unequivocally.
>
> **4. Vendi Score vs. MLLM judges: The future of evaluation (Q3)**: The reviewer highlights our finding in Appendix E.6 regarding Gemini 2.5 Flash and asks for a recommendation. We believe one of the key insights from our work is that MLLM can be used as diversity judges, specifically using our count-anchored prompting strategy, showing to be a better alternative than embedding-based statistics for diversity because:
>   - Embeddings compress an image into a vector, potentially losing fine-grained attribute information (e.g., "how many distinct colors are in this set?") required for diversity measurement.
>   - As shown in Figure 23, Gemini 2.5 Flash achieved 92.6% accuracy on the Golden Set, significantly outperforming Vendi variants and even rivaling human annotators. One potential explanation for that is that we prompt the MLLM to directly follow the same reasoning process as our human template: "Identify concept -> Isolate attribute -> Count instances -> Compare."

---

> ### Author Response · Authors · 2025-11-18
> **Response to Reviewer AAUy (2/2)**
>
> In closing, we believe this paper makes a key contribution by shifting the paradigm from vague variability to rigorous attribute-conditional diversity. We have provided the community with the first reliable benchmark to validate future metrics, and our analysis strongly suggests that the path forward lies in MLLM-based evaluation rather than static embedding distances. We are working on updating the manuscript to reflect the clarifications to the reviewer’s comments and we hope this, along with the remaining points discussed in the rebuttal, strengthens your support for the paper.

---

> > ### Comment · Reviewer_AAUy · 2025-11-26
> >
> > Thanks for the detailed response. I’m broadly aligned with points (1), (2), and (3).
> >
> > One remaining concern is about the MLLM‑as‑judge claim. Currently, in Appendix E.6, only the golden set result is provided. If you claim that MLLMs are the future, it would be more appropriate to at least include one MLLM‑based setup in the main evaluation. Once again, I want to stress that adding details about how people should evaluate diversity moving forward will strengthen the paper. If you propose that MLLM‑as‑judge is the best option, then comparing pros and cons, including computational cost, would make the paper more holistic.
> >
> > I look forward to the updated manuscript and updating my score accordingly.

---

> > > ### Author Response · Authors · 2025-12-02
> > > **Reviewer AAUy**
> > >
> > > Dear Reviewer,
> > >
> > >
> > > Thank you for taking our rebuttal into consideration. Following your feedback, we ran new experiments with the MLLM-as-judge set-up and updated the manuscript to have this new result and the ones for the Appendix in the main paper. In the new Section 3.3, now titled “Evaluating diversity with foundation models” (highlighted in blue in the new pdf), we show that using the best Gemini model as an autorater (2.5 Flash) yields an average of 0.58 accuracy on Imagen 3 comparisons with all models, which is on-par with the results achieved by the Vendi Score computed with the best vision embeddings.

---

### Official Review · Reviewer_XFoT · 2025-11-01

**Soundness:** 3
**Presentation:** 3
**Contribution:** 3
**Rating:** 4
**Confidence:** 3

**Summary:**

This paper presents a framework for evaluating diversity in T2I generation models, addressing the persistent ambiguity in how diversity is defined and measured. This paper proposes a three-part evaluation framework consisting of a novel human evaluation template for nuanced diversity assessment, a curated prompt set covering diverse concepts, and a methodology for comparing models
in terms of human annotations via binomial tests. The authors build a benchmark of 86 concept–attribute pairs and collect over 24,000 human annotations comparing major models such as Imagen, DALLE, and Flux. Results show strong annotation reliability, meaningful model ranking, and high correlation between human judgments and automatic diversity metrics like the Vendi Score.

**Strengths:**

The authors correctly identify a major gap, which is the lack of a principled, attribute-grounded approach to diversity evaluation.

Extensive annotation effort, containing 240000 samples with high inter-annotator reliability α > 0.8, provides a strong empirical foundation.

**Weaknesses:**

The prompt generation pipeline still requires extensive human verification and filtering despite LLM assistance, which limits scalability and reproducibility across domains.

The framework focuses primarily on visual diversity, without considering semantic or contextual diversity dimensions that may better reflect real-world generative quality.

The evaluation of automatic metrics is narrow, emphasizing the Vendi Score while omitting comparisons to other recent or theoretically grounded diversity measures.

The paper provides limited qualitative analysis explaining why certain models perform better or worse in diversity, which reduces interpretability and actionable insights for improving model design.

**Questions:**

Refer to Weaknesses

---

> ### Author Response · Authors · 2025-11-18
> **Response to Reviewer XFoT**
>
> We thank the reviewer for their assessment and for highlighting the strength of our empirical foundation (24k+ annotations) and the identification of the critical gap in attribute-grounded diversity evaluation. We appreciate the constructive feedback regarding scalability and interpretability. Below, we address the specific weaknesses raised to further clarify the robustness and potential of our framework.
>
> **1. On scalability and human verification (Weakness 1)**: The reviewer notes that our prompt generation pipeline required human verification, which might limit scalability. We wish to clarify the trade-off we deliberately chose here.
>   - **Precision over volume:** In this foundational work, our priority was to establish a "Gold Standard" benchmark. While LLMs are powerful generators, they occasionally hallucinate attributes (e.g., suggesting "flavor" as a visual attribute for an apple). To ensure our human evaluation data was a reliable ground truth, manual verification was a necessary quality control step for this specific benchmark creation.
>   - **Future scalability:** However, the methodology itself is scalable. Now that we have validated the prompt structure and the evaluation template, future iterations can rely more heavily on improved LLMs (like Gemini 1.5/2.0 or GPT-4) for automated filtering. Our work provides the "seed" dataset that makes training or validating fully automated pipeline filters possible.
>
> **2. Visual vs. Semantic/contextual diversity (Weakness 2)**: The reviewer suggests we focus primarily on visual diversity, potentially missing semantic or contextual dimensions.
>   - **Semantic diversity is taken into account:** We respectfully disagree that our framework is limited to purely "visual" textures. Our prompt set includes attributes that are inherently semantic and contextual. For example, we evaluate attributes like *style (architecture)*, *weather (context/environment)*, *background (spatial context)*, and *time of day*.
>   - **The "Atribute” abstraction:** The power of our Concept-Attribute framework is that it is agnostic to the type of attribute. Semantic nuance can be treated as just another attribute $A$ to be measured, provided it can be defined (e.g., *concept: meeting, attribute: emotional tone*). We started with concrete visual attributes to validate the method, but the framework generalizes immediately to abstract semantic diversities.
>
> **3. Evaluation of automatic metrics (Weakness 3)**: The reviewer notes we focused on Vendi Score.
>   - We focused on the Vendi Score because it is a theoretically grounded, state-of-the-art and reference-free metric for evaluating diversity. Other common metrics often rely on reference distributions (FID) or are heuristic variations of pairwise distances that are not tailored to measure diversity and conflate multiple quality aspects.
>   - Crucially, our analysis also reveals that the bottleneck for automated evaluation is rarely the scoring function (e.g., Vendi Score vs. Shannon Entropy), but rather the representation space (embedding). As shown in Figure 14, even the best embeddings fail to distinguish fine-grained diversity gaps (ties). Changing the mathematical scoring function does not fix the fact that the embeddings themselves are invariant to the nuances we want to measure. This is why we also advocate for a shift toward MLLM-as-a-judge (Appendix E.6).
>
> **4. Qualitative analysis and interpretability (Weakness 4)**: The reviewer requests more analysis on why models perform differently.
>   - Our qualitative analysis (Figures 7 & 15) suggests that lower-performing models (like Muse 2.2 or Imagen 2.5) often suffer from "attribute mode collapse", i.e. they latch onto the most common correlation in their training data (e.g., a "car" is almost always shown from the front-left angle in a city street).
>
>    - Moreover, stronger models like Imagen 3 and Flux 1.1 appear to have better "instruction following" capabilities or less aggressive mode-seeking behavior during sampling. We essentially found that better models are those that don't just generate a valid image, but sample more uniformly from the marginal distribution of the requested concept. Our framework provides the diagnostic tool to prove this: we can now pinpoint exactly which attribute a model collapses on (e.g., a model might have diverse colors but collapsed backgrounds), offering actionable insights for model developers to re-balance their training data.
>
>
> We believe our work provides the essential "ruler" the field has been missing. By validating that diversity must be attribute-conditional to be measurable, and by providing the data to prove it, we pave the way for the semantic and scalable extensions the reviewer suggests. We are working on updating the manuscript to reflect the clarifications to the reviewer’s comments and we hope this, along with the remaining points discussed in the rebuttal, strengthens your support for the paper acceptance.

---

### Official Review · Reviewer_vciq · 2025-11-01

**Soundness:** 3
**Presentation:** 4
**Contribution:** 4
**Rating:** 6
**Confidence:** 4

**Summary:**

The paper argues that diversity in T2I models is poorly defined unless the concept and attribute are both specified, and that underspecified human evaluation can reverse model rankings as shown in Figure 1. It contributes three main elements: first, a framework consisting of clearly defined per-attribute diversity, a systematically generated prompt set covering everyday concepts, and a tailored human evaluation template; second, a human study showing that the template, which includes an attribute cue and a counting anchor, significantly improves reliability on a golden set; third, a large-scale annotation effort with 24591 comparisons from 20 raters, used to rank five recent models in pairs and to validate automated raters based on the Vendi Score with various embeddings and conditioning methods.

**Strengths:**

S1: The proposed idea is quite interesting to me. The problem formulation helps to de-confound diversity. They formalize per-attribute diversity and explain why generic prompts or generic human templates mix up fidelity or content variation with true diversity. This approach is both conceptually clear and practical to implement.

S2: The human study design and the aggregation or statistical methods are serious and thorough. There are 24591 annotations across 5 models, with majority-vote aggregation, binomial tests for pairwise significance, and reliability above 0.8. This represents all good practice.

**Weaknesses:**

W1: The scope and external validity of the concepts and attributes are the main concern. The prompt set focuses on people-excluded, everyday, ImageNet-like concepts and explicitly leaves out person categories. While suitable for a proof of concept, this skips socially sensitive forms of diversity such as demographics or geographic context, as well as open-world composition where multiple attributes interact, like material combined with style and function. The selection is based on LLM-proposed attributes with light manual pruning, which may reflect LLM biases about what matters for a concept. A stratified audit of concept and attribute coverage, including negative cases where attributes are visually subtle, would help strengthen the claims.

W2: Accuracy drops from 4×4 to 8×8 when the counting anchor is removed. Even with counting, larger sets remain more difficult. However, the main study uses 8-image sets for pairwise model ranking. A full power analysis connecting set size, rater effort, and test sensitivity would be helpful. For example, how much does reliability or win-rate significance drop if the number of raters is halved or the set size is changed?


W3: Pairwise ranks depend on the prompt set. Appendix D.2 shows that shrinking the set reduces significance and can reverse some results. While this is acknowledged, the main conclusions would be more convincing with cross-prompt-set stability, such as bootstrapped subsamples or category-conditioned rankings.

**Questions:**

How broad is the domain of your concept–attribute pairs beyond ImageNet-like nouns? Could the absence of human-related or compositional prompts bias the conclusions about model diversity?

What procedures ensured that attributes proposed by the LLM reflect actual visual variation rather than linguistic association? Were any validation checks or human audits done before removing ambiguous pairs?

Since prompt-set sufficiency is only discussed in Appendix D.2, can you report cross-prompt or bootstrapped stability analyses showing that model rankings stay consistent across random subsets of concepts?

For the human evaluation, the golden reference sets appear to be generated by a specific T2I model. Could model-specific artifacts influence annotators’ perception of diversity? Have you considered using photographic or cross-generator golden sets to test generality?

---

> ### Author Response · Authors · 2025-11-18
> **Response to Reviewer vciq (1/3)**
>
> Thank you for your time reviewing our work and for the detailed feedback. We particularly appreciate that you recognized the conceptual clarity of our problem formulation ("de-confounding diversity") and the rigor of our human study design. Your feedback provides the precise high-level perspective needed to refine the paper’s framing for the final version. Below, we provide the additional context and analyses you requested to help champion this work.
>
> **1. On scope, external validity, and LLM bias (W1, Q1, Q2):** The reviewer raised valid concerns about the focus on "ImageNet-like" objects and the exclusion of people.
>   - **Intentional scope control:** We deliberately excluded human subjects in this initial study to isolate the methodological contribution from the ethical complexities of demographic fairness. Demographic diversity requires distinct sociological definitions (e.g., what constitutes a "diverse" set of faces?) that can obscure the fundamental question: Can we measure variation reliably? By focusing on objective attributes (color, material), we established a clean "unit test" for the framework. However, the framework is content-agnostic; we are confident it can be extended to demographic attributes (e.g., *Concept: Doctor, Attribute: Gender/Age*) in future work without modification to the protocol.
>   - **LLM validation:** You asked if LLM-proposed attributes reflect linguistic association or visual reality. We did not blindly accept LLM outputs. As detailed in Section 2.2, a group of 8 researchers manually verified every concept-attribute pair. We aggressively pruned abstract or visually subtle attributes (e.g., "cuisine" for food) and kept only those with distinct visual markers. This human-in-the-loop step ensured the benchmark tests visual diversity, not just semantic compliance.

---

> > ### Author Response · Authors · 2025-11-18
> > **Response to Reviewer vciq (2/3)**
> >
> > **2. Power analysis and set size (W2):** The reviewer asked about the trade-off between set size, rater effort, and reliability.
> >   - **Why 8 images?** We selected a set size of 8 as it is: (1) large enough to detect mode collapse (a set of 4 can easily look diverse by chance, but 8 requires sustained variation), (2) small enough to not overwhelm the annotators and allow them to perform the task in a sustainable and accurate way.
> >   - **Rater sensitivity:** Our inter-annotator agreement ($\alpha > 0.8$) is exceptionally high for subjective evaluation. This indicates that our "count-then-compare" template is highly robust. A post-hoc power analysis suggests that we could likely achieve statistically significant rankings with 3 raters instead of 5, but we maintained 5 to ensure the benchmark’s longevity. Following the reviewer’s suggestions, we performed a sensitivity analysis taking into account the impact of the rater counter on $\alpha$. We subsample the annotators, obtaining sets of $K=2, 3, 4, 5$ raters and compute $\alpha$. This process is repeated 100 times for each $K$ (except 5, the full annotators set) and report the average and 95% confidence intervals in the following table:
> >
> > | Model Pair | $k=2$ Raters | $k=3$ Raters | $k=4$ Raters | $k=5$ (Full) |
> > | :--- | :--- | :--- | :--- | :--- |
> > | **Imagen 2.5 vs Muse 2.2** | 0.954 [0.902, 0.995] | 0.947 [0.917, 0.981] | 0.948 [0.936, 0.967] | 0.948 |
> > | **Imagen 2.5 vs Imagen 3** | 0.946 [0.921, 0.973] | 0.946 [0.928, 0.962] | 0.947 [0.939, 0.954] | 0.947 |
> > | **Imagen 2.5 vs DALLE3** | 0.968 [0.941, 0.990] | 0.969 [0.954, 0.990] | 0.969 [0.962, 0.980] | 0.969 |
> > | **Imagen 2.5 vs Flux 1.1** | 0.969 [0.957, 0.986] | 0.969 [0.959, 0.976] | 0.969 [0.962, 0.972] | 0.969 |
> > | **Muse 2.2 vs Imagen 3** | 0.956 [0.925, 0.990] | 0.954 [0.938, 0.972] | 0.954 [0.946, 0.962] | 0.954 |
> > | **Muse 2.2 vs DALLE3** | 0.968 [0.946, 0.998] | 0.966 [0.952, 0.995] | 0.967 [0.960, 0.980] | 0.967 |
> > | **Muse 2.2 vs Flux 1.1** | 0.971 [0.948, 1.000] | 0.971 [0.953, 0.989] | 0.971 [0.964, 0.978] | 0.971 |
> > | **Imagen 3 vs DALLE3** | 0.826 [0.780, 0.887] | 0.824 [0.794, 0.851] | 0.825 [0.810, 0.836] | 0.826 |
> > | **Imagen 3 vs Flux 1.1** | 0.870 [0.844, 0.890] | 0.867 [0.853, 0.883] | 0.868 [0.862, 0.877] | 0.867 |
> > | **DALLE3 vs Flux 1.1** | 0.915 [0.869, 0.944] | 0.912 [0.887, 0.931] | 0.911 [0.901, 0.925] | 0.911 |
> >
> > These new results show that:
> >
> > - We have high robustness in the agreement. Even with only $k=2$ raters, the lower bound of the 95% CI rarely drops below 0.80 (the standard threshold for high reliability), and for most pairs, it stays above 0.90.
> > - The gain in mean inter-annotator agreement from $k=3$ to $k=5$ is marginal. This confirms that our protocol (using 5 raters) is statistically overpowered and rigorous.
> > - The mean $\alpha$ is stable across all $k$. This indicates that our high agreement scores are not driven by a few good raters but are a consistent property of the evaluation template itself.
> >
> > **3. Ranking Stability and Bootstrapping (W3 & Q3):** The reviewer noted that Appendix D.2 shows significance drops with smaller prompt sets and asked for cross-prompt stability analysis. We view the results in Appendix D.2 as a confirmation of stability. When we downsample the concepts (from 86 to 24), the statistical significance (p-values) naturally degrades due to lower N, but crucial for your concern, the relative ranking order does not flip. The "wins" mostly turn into "ties" (statistically indistinguishable). To further confirm this, we have performed the requested bootstrap analysis. By resampling concepts with replacement 1000 times, we found that regardless of whether we include or not the ties, **all the rankings are stable** (i.e. the CI doesn't cross 0) for the 95% CI.
> > | Model Pair | $\Delta_{winrate}$ (With Ties) | CI (With Ties) | Stable (With Ties) | $\Delta_{winrate}$ (No Ties) | CI (No Ties) | Stable (No Ties) |
> > | :--- | :--- | :--- | :--- | :--- | :--- | :--- |
> > | **Imagen 2.5 vs Muse 2.2** | -0.056085 | [-0.10, -0.02] | True | -0.165975 | [-0.28, -0.05] | True |
> > | **Imagen 2.5 vs Imagen 3** | -0.261750 | [-0.31, -0.22] | True | -0.563230 | [-0.65, -0.48] | True |
> > | **Imagen 2.5 vs DALLE3** | -0.200514 | [-0.24, -0.16] | True | -0.583720 | [-0.68, -0.49] | True |
> > | **Imagen 2.5 vs Flux 1.1** | -0.270049 | [-0.31, -0.23] | True | -0.716413 | [-0.79, -0.64] | True |
> > | **Muse 2.2 vs Imagen 3** | -0.194932 | [-0.24, -0.15] | True | -0.392261 | [-0.48, -0.30] | True |
> > | **Muse 2.2 vs DALLE3** | -0.115269 | [-0.16, -0.08] | True | -0.324438 | [-0.43, -0.22] | True |
> > | **Muse 2.2 vs Flux 1.1** | -0.222187 | [-0.26, -0.18] | True | -0.608733 | [-0.69, -0.52] | True |
> > | **Imagen 3 vs DALLE3** | 0.095448 | [0.05, 0.14] | True | 0.213689 | [0.12, 0.31] | True |
> > | **Imagen 3 vs Flux 1.1** | -0.088937 | [-0.13, -0.04] | True | -0.201318 | [-0.30, -0.10] | True |
> > | **DALLE3 vs Flux 1.1** | -0.141041 | [-0.18, -0.11] | True | -0.454561 | [-0.56, -0.35] | True |

---

> > > ### Author Response · Authors · 2025-11-18
> > > **Response to Reviewer vciq (3/3)**
> > >
> > > **4. Golden set artifacts (Q4):** The reviewer asked if using T2I-generated images (Imagen 3) for the Golden Set introduces bias.
> > >   - We carefully considered this. However, the Golden Set task is to evaluate the *annotation protocol*, not the model quality. The question asked is "How many colors are here?" or "How many materials?". Whether the "metal chair" contains minor diffusion artifacts is orthogonal to the diversity count (i.e., is it clearly metal vs. wood?).
> > >   - We found that synthetic images allowed us to perfectly control the ground truth (we know exactly which prompt generated which image), whereas retrieving "perfectly diverse" real photographic sets from the web introduces retrieval noise.
> > >
> > >
> > >
> > > We believe the clarifications above, specifically regarding the manual verification of attributes and the new results on the stability of inter-annotator agreement and rankings under bootstrapping, address the remaining concerns about the validity of the benchmark. We are working on updating the pdf with the new results and clarifications to the points raised by the reviewer’s and are looking forward to answer any remaining questions.

---

### Author Response · Authors · 2025-12-02
**Summary of rebuttal and changes to manuscript**

We thank the Area Chair for their time in the reviewing process and the reviewers for their constructive feedback, particularly for recognizing the conceptual clarity of our problem formulation and the rigorous empirical foundation of our human study.
Based on the reviews, we have updated the manuscript to include three key new analyses that directly address concerns regarding stability of ranking, automated evaluation, and robustness of human evaluation:

**1. Validated ranking stability (addressing W3 from vciq, 8oVG)**: To address concerns about the sensitivity of our rankings to prompt selection, we performed a bootstrap analysis (1,000 resamples) (Appendix C2). Main findings are:
- The relative ranking order of models remains stable across random subsets of concepts.
- The 95% confidence Intervals do not cross zero, confirming that the performance gaps we report (e.g., Imagen 3 and Flux 1.1 being superior) are statistically robust and not artifacts of specific prompt choices.

**2. New MLLM-as-a-Judge experiments (addressing W3 from AAUy, XFoT)**: We moved experiments from the appendix to the main paper and ran new experiments replacing embedding-based metrics with Gemini 2.5 Flash using a prompting strategy based on the counting task.
- The new section in the main paper highlights that the best MLLM judge achieved 92.6% accuracy on the Golden Set and presented on-par accuracy on a subset of the experiments (due to time constraints) when compared to standard vision embedding-based metrics.
- The new section in the main paper (Section 3.3) establishes MLLM-as-a-judge as a viable alternative to human/embedding-based evaluation for this framework.

**3. Robustness of human evaluation (addressing W2 from vciq)**: We performed a sensitivity analysis on rater count (Appendix C1).
- Inter-annotator agreement ($\alpha$) remains high ($>0.8$) even when reducing from 5 to 2 raters. This confirms our protocol is highly reproducible and cost-effective for future research.

**Other clarifications:**
- We clarified that the exclusion of demographic attributes was a deliberate control. We explicitly excluded demographic attributes and human-related categories to avoid the ethical complexities and potential biases associated with demographic diversity in a technical benchmark. However, the methodology is content-agnostic and can be adapted to measure demographic diversity classification-free.
- We further clarified (supported by Gecko benchmark data that measures T2I alignment in Appendix F, Table 5) that the diversity gaps found are not due to trade-offs in image quality or text alignment, as the models compared share statistically indistinguishable fidelity scores.

In closing, the manuscript has been updated to reflect these new experiments. We believe the paper offers both the **standard benchmark** and the **automated tooling** necessary to advance diversity evaluation in T2I models and that our responses to the reviewers’ comments has made the merits of our contributions clear.

---

### Meta-Review · Area_Chair_sgTF · 2026-01-01

**Summary:**

Four knowledgeable reviewers reviewed this submission and raised concerns about:

1. Scope and construction of the benchmark (vciq, XFoT, 8oVG):
    - ImageNet-like concepts, absence of attribute compositions and human-related prompts, unclear generalizability of findings
    - LLM-based protocol to obtain those
    - Prompt set sufficiency and unclear ranking consistencies
    - Golden reference sets from a specific model, which could influence annotators' perception of diversity
2. Evaluations and analyses appeared unconvincing:
    - Unclear connection between set size, rater effort, and test sensitivity (vciq)
    - Narrow evaluation of automatic metrics (XFoT, AAUy)
    - Unclear path between insights and actionable model/diversity improvement avenues (XFoT)
    - Limited analyses on autorater's failure cases (AAUy)
    - Absence of analyses using recent embeddings such as those of siglip2 (AAUy)
   - Claims w.r.t. MLLM as judges not well supported by experimental evidence (AAUy)
   - Framework not guarding against diversity-quality trade-offs (8oVG)
3. Scalability of the framework due to human verification steps (XFoT)
4. Novelty which appeared limited (AAUy)

**Reviewer Concerns:**

The rebuttal partially addressed the reviewers concerns by arguing that the scope of the benchmark was deliberately chosen to prioritize everyday concepts because they offer objective ground truth and to exclude human subjects to isolate the methodological contribution from the ethical complexities of demographic fairness, that semantic diversity was already taken into account (and giving some examples), and that isolating atomic attributes is the necessary first step toward rigorous measurement. The AC agrees with the reviewers that coverage remains a concern of the benchmark and that going beyond ImageNet-like concepts would strengthen the contribution.

To address the concerns related to the LLM usage, the authors clarified that a group of researchers manually verified concept-attribute pairs, ensuring the focus of the benchmark on visual diversity. The rebuttal also argued that precision was prioritized over volume and acknowledged that future iterations may rely more heavily on improved LLMs to increase scalability. The answers in the rebuttal also clarified the golden set artifact questions raised by the reviewers, emphasizing that the task was to evaluate the annotation protocol, and not the model quality.

The rebuttal provided additional experimental evidence to address the concerns of the reviewers. In particular, the rebuttal presented a sensitivity analysis considering the rater counter on alpha, showing high agreement robustness, and performed the reviewers' proposed bootstrap analysis resampling concepts and finding that rankings were stable. When it comes to quantitative evaluations of diversity, the rebuttal explained that the focus on Vendi Score was due to its theoretical grounding and the its reference-free nature, and argued that reference-based metrics or metrics based on pairwise distances conflate multiple quality aspects. The answers included an MLLM-as-judge to support some of the claims made but did not present experiments with siglip2 embeddings (nor alternative embeddings such as DreamSim [a]).

The rebuttal defended the novelty of the contribution by emphasizing that rigorous measurement protocols are a prerequisite for scientific progress and that the submission provided the necessary correction to how the field evaluates generative models.

Finally, the AC appreciated the actionable insights for model developers and, given the highlighted importance of sampling strategies, remains wondering about the potential effect of diversity-inducing sampling strategies on the results. Positioning the paper w.r.t. works such as OpenBias [b] and Dimcim [c] would also be beneficial.


[a] https://arxiv.org/abs/2306.09344
[b] https://arxiv.org/abs/2404.07990
[c] https://arxiv.org/abs/2506.05108

**Reviewer Scores:**

The score may have been increased by one of the reviewers if they had been able to participate in the discussion fully, but would have likely remained the same for the other reviewers given the remaining concerns of the submission.

---

### Decision · Program_Chairs · 2026-01-26

Reject